# Stable representation of a naturalistic movie emerges from episodic activity with gain variability

Ji Xia [1✉], Tyler D. Marks [2], Michael J. Goard [2,3,4,5] & Ralf Wessel[1,5]

Visual cortical responses are known to be highly variable across trials within an experimental session. However, the long-term stability of visual cortical responses is poorly understood. Here using chronic imaging of V1 in mice we show that neural responses to repeated natural movie clips are unstable across weeks. Individual neuronal responses consist of sparse episodic activity which are stable in time but unstable in gain across weeks. Further, we find that the individual episode, instead of neuron, serves as the basic unit of the week-to-week fluctuation. To investigate how population activity encodes the stimulus, we extract a stable one-dimensional representation of the time in the natural movie, using an unsupervised method. Most week-to-week fluctuation is perpendicular to the stimulus encoding direction, thus leaving the stimulus representation largely unaffected. We propose that precise episodic activity with coordinated gain changes are keys to maintain a stable stimulus representation in V1.

[1] Department of Physics, Washington University in St. Louis, St. Louis, MO, USA. [2] Neuroscience Research Institute, University of California, Santa Barbara, CA, USA. [3] Department of Molecular, Cellular, and Developmental Biology, University of California, Santa Barbara, CA, USA. [4] Department of Psychological & Brain Sciences, University of California, Santa Barbara, CA, USA. [5] These authors jointly supervised this work: Michael J. Goard and Ralf Wessel.
✉email: xiaji@wustl.edu

Stimulus-driven activity is highly variable across repeated trials within a recording session[1–5]. Furthermore, in chronic recordings covering multiple stimulus sessions, session-to-session fluctuation tends to be qualitatively different from trial-to-trial variability within sessions[6–9]. Even without learning, the same neuron population responds unstably under the same environmental and behavioral conditions across days[10–14]. However, not all the brain areas share the same instability[9]. For example, neural activity from posterior parietal cortex[11], hippocampus[14], and primary olfactory cortex[15] exhibit large changes across days, while HVC (proper name) neural activity remains stable in long-term recordings[16].

How does stimulus-driven activity in V1 change across days under a nominally constant condition? Recently, several studies shed light on how V1 stimulus-driven activity changes in the long term in responses to drifting gratings[17–19]. Even though day-to-day variations were larger than trial-to-trial variations[18], stable tuning over weeks was found in most tuned neurons[17,18]. Yet few reported on the long-term stability of neural responses to natural movies[19,20]. Natural movie responses are sparser and more precise than neural responses to artificial stimuli such as drifting gratings[21,22]. Moreover, responses to natural stimuli cannot be predicted from responses to drifting gratings[23,24]. Thus, the long-term stability of neural responses to natural movies is not necessarily the same as that to drifting gratings. Indeed, data from our group showed that single neural responses to natural movies were significantly more unstable than drifting grating responses[25].

This session-to-session fluctuation raises an important question: Is there a stable representation of natural stimuli hidden in the unstable neural activity in V1? Stable stimulus representation is possible when neural fluctuations reside in a space orthogonal to the stimulus encoding dimensions[26]. Intuitively, if one neuron's session-to-session fluctuation affected the encoding of stimulus, then the other neurons' fluctuation could compensate for its influence. Moreover, the stimulus could be encoded in a low-dimensional subspace of the high-dimensional population activity[27,28]. In that case, the random fluctuation in the high-dimensional neural space would likely be perpendicular to the low-dimensional subspace of stimulus encoding, often referred to as the stimulus encoding dimension. Clarification of these possibilities requires long-term recordings in response to repeated stimulation, identification of the stimulus encoding dimensions, and quantification of neural fluctuation within the high-dimensional population activity.

To address the question of stable stimulus representation in unstable neural activity, we analyzed a dataset from longitudinal two-photon calcium imaging of excitatory neurons in the primary visual cortex of awake, head-fixed mice during visual stimulation with repeated identical natural movie clips across weeks. We found that single neural responses consisted of episodic activity that was precise in time during the natural movie across weeks. However, firing rates during those spiking episodes were unstable across weeks. Moreover, within the same neuron, firing rates of different spiking episodes varied in distinct temporal patterns across weeks. By fitting a linear model, we found that episodic activity was the basic unit of the week-to-week fluctuation. Importantly, despite the unstable episodic activity, we extracted a low-dimensional stable representation of time in the natural movie from neuronal population activity across weeks. We propose that precise episodic activity with coordinated gain changes are keys to maintain a stable stimulus representation in V1.

## Results

### Single neuron responses to natural movies are unstable across weeks.

To investigate the long-term variability of cortical responses, we used a dataset that consisted of chronic GCaMP6s imaging of excitatory neurons in V1 L2/3 of awake, head-fixed mice during visual stimulation with repeated natural movies (30 trials per session; one session per 7 ± 1 days; over 5–7 weeks) (Fig. 1a)[25]. Single neuron responses varied in a largely stochastic manner across trials within a recording session (week) as described before[1–3] and, importantly, varied in a qualitatively different manner across weeks (Fig. 1b). We quantified this response variation across weeks in terms of the "similarity", defined as the correlation coefficient between trial-averaged neural responses (within a week) for a given neuron between pairs of weeks and averaged across all neurons. Similarity largely decreased over time using the first week of recording as the reference (Fig. 1c). Specifically, the similarities of the fifth week were significantly lower than the similarities of the second week (one-sided Wilcoxon signed-rank test, $p = 0.0035$, ten imaging fields). In a complementary analysis, to compare how single neuronal activity varied across weeks, we computed the difference of trial-averaged activity across weeks (Supplementary Fig. 1a). The change of trial-averaged $\Delta F/F$ across weeks was significantly higher than baseline variability within a week (Supplementary Fig. 1b; two-sided Mann−Whitney U test, $p = 0.0029$, ten imaging fields). In conclusion, consistent with an earlier study[25], but using complementary analyses, we showed that single neuron responses to natural movies are unstable across weeks.

### Single neuron responses consist of episodic activity with distinct episode-specific rate variations across weeks.

The episodic nature of cortical neuron responses to naturalistic visual stimuli (Fig. 1b)[21,29–31] provides the unique opportunity to study neural variability with respect to episodic spiking. Neurons in the visual cortex are known to respond to naturalistic movies sparsely with temporally precise, but stochastic, spiking within a few well-timed "spiking episodes"[21,22,32]. Is the change in single neuron spiking across weeks (Fig. 1c and Supplementary Fig. 1) dominated by changes in spike timing or by changes in spike counts? To address this question, we inferred spiking activity[33] and defined spiking episodes (Fig. 2a; see "Methods") based on peaks in the smoothed peristimulus time histogram (PSTH). Note that the inferred spiking activity might correspond to bursts of spikes instead of a single spike due to limitations of calcium imaging[34]. A neuron usually possessed multiple spiking episodes and episodes from different neurons overlapped (Fig. 2b). To quantify the precision of spiking episodes across weeks, we computed the durations of spiking episodes (Fig. 2c). The right-skewed distribution of durations showed that most of the spiking episodes had short durations (median duration: 0.66 ± 0.17 s, ten imaging fields). Furthermore, compared with spiking episodes defined based on PSTH within weeks, the median of durations of spiking episodes defined based on PSTH across weeks only increased by at most 2 time steps (0.2 s) for each imaging field (Fig. 2c, the median duration of spiking episodes based on PSTH within weeks: 0.59 ± 0.074 s, ten imaging fields). The short durations of spiking episodes and a small increase compared with data within weeks indicated that episodic activity had rather precise and stable timing across weeks. In contrast, the inferred spike rates during those spiking episodes changed more from trial-to-trial across weeks than that within each week (Fig. 2d). Importantly, inferred spike rates during each episode for a given neuron varied in different patterns from week to week (Fig. 2e). The diverse inferred spike rate variation for different spiking episodes raised the question whether inferred spike rates during spiking episodes within the same neuron change independently across weeks. We quantified the similarity between inferred spike rate variability

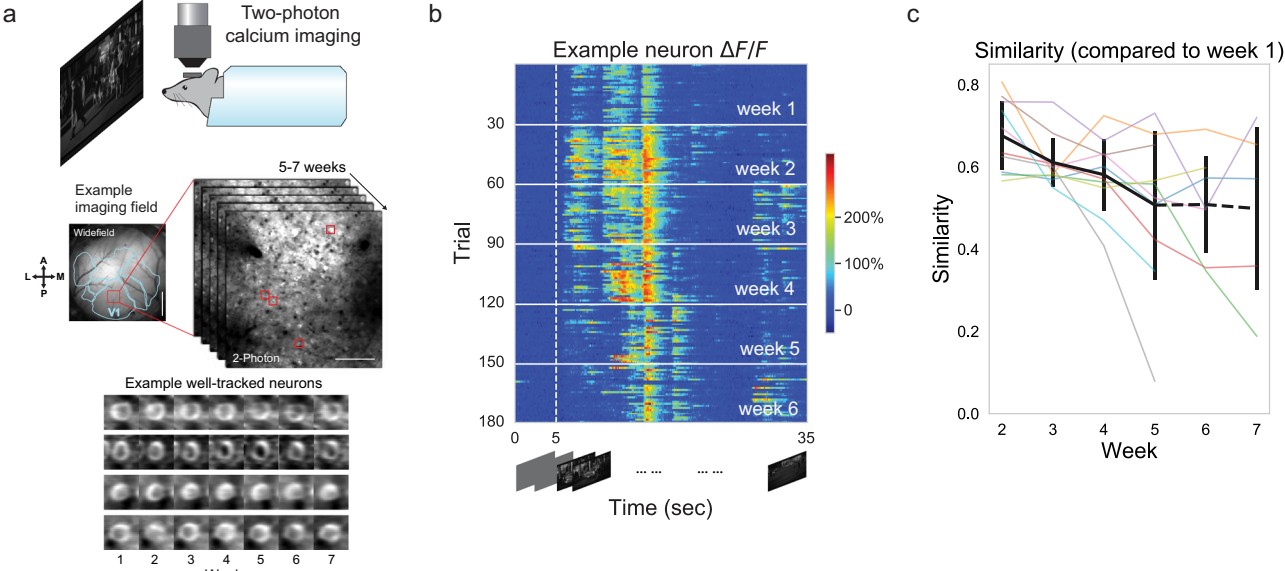

**Fig. 1 Single neuron responses to natural movies are unstable across weeks. a** Experimental setup. We performed chronic calcium imaging of excitatory neurons in the primary visual cortex of awake, head-fixed mice during visual stimulation with repeated natural movies. The visual cortex (contralateral to visual stimulus delivery) is retinotopically mapped in Emx1-Cre::TITL-GCaMP6s mice. V1 fields are chosen from the region selective for the center of the presentation screen. Widefield scale bar $= 1$ mm; two-photon scale bar $= 100$ μm. The average activity of four example well-tracked neurons across weeks are shown in the bottom panel. **b** $\Delta F/F$ responses of one example neuron during the same natural movie clip for 30 trials per experimental session for 6 weeks (movie starts at 5 s and lasts for 30 s duration). We recorded one experimental session per week. **c** Similarity (correlation coefficient between trial-averaged $\Delta F/F$) averaged over neurons during week 1 and that during other weeks are plotted for all the recorded imaging fields. Different imaging fields are denoted by different colors. The black curve with the error bar denotes the mean and standard deviation of similarity over imaging fields. Ten imaging fields have recordings for weeks 1—5. Only a subset of imaging fields has recordings on week 6 (seven fields) and week 7 (five fields). Specifically, the similarities of the fifth week were significantly lower than the similarities of the second week (one-sided Wilcoxon signed-rank test, $p = 0.0035$, ten imaging fields).

during different spiking episodes as the mean correlation coefficient between mean inferred spike rate across weeks (Fig. 2e). For most neurons, the similarity of inferred spike rate changing patterns across spiking episodes was low, although significantly higher than the chance level (Fig. 2f, one-sided Mann−Whitney U test, $p = 2.45 \times 10^{-42}$, 1404 neurons with more than 1 spiking episodes). This means that different spiking episodes within the same neuron have different, but not completely independent, inferred spike rate variations across weeks. Moreover, the similarity between inferred spike rate changing patterns was significantly lower than that expected from i.i.d. Poisson statistics (Fig. 2f, one-sided Mann−Whitney U test, $p = 1.03 \times 10^{-143}$, 1404 neurons with more than 1 spiking episodes). Consequently, assuming spike trains of all the trials were independent Poisson spike trains, the inferred spike rates of distinct spiking episodes within the same neuron followed significantly different variations across weeks. The difference in inferred spike rate changing patterns of spiking episodes within the same neuron suggests that the basic unit of the week-to-week fluctuation is the spiking episode instead of neuron.

**Latent factors resembling episodic activity with gain changes capture the across-week fluctuations.** To identify the basic unit of the week-to-week fluctuation in an unbiased fashion, we switched from single-neuron analysis (Figs. 1, 2) to population analysis (Fig. 3 and Supplementary Figs. 2, 3), thus including the potential impact of coordinated activity. We decomposed population activity into latent factors that can have independent gain changes across trials. For this purpose, we chose the recently introduced tensor component analysis (TCA)[3,35], which provides an unsupervised way to identify latent factors of the recorded population activity. Specifically, we organized neuronal responses

into a three-dimensional tensor (neuron × time × trials) and decomposed this tensor into $R$ components, each consisting of a neuron factor, a temporal factor, and a trial factor (Fig. 3a). Thus, TCA achieves a simultaneous, interlocked dimensionality reduction across neurons, time, and trials. For each component, (i) the neuron factor indicates how the component is shared across neurons, (ii) the temporal factor reflects the component's temporal profile on every trial, and (iii) the trial factor enumerates how the component's gain changes across trials. Within this framework, a neuronal response can be approximated by the reconstructed response, which is a linear combination of these TCA components (Fig. 3b). As TCA components mainly capture correlated activity across neurons or trials[35], the reconstructed responses from TCA components can be viewed as denoised responses, i.e., the responses from which independent noise has been removed.

Within this unsupervised TCA dimensionality reduction method (Fig. 3c), the pronounced peaks in the temporal factors (Fig. 3c, center) revealed shared episodic activity across neurons (Fig. 2a, b). Importantly, the distribution of the temporal factors across all 40 TCA components (Fig. 3c, center) revealed the scattering of the episodic activity across the duration of a trial (Fig. 2b). For a given TCA component (of the chosen $R = 40$ components), the neurons with a high neuron factor value (Fig. 3c, left) had episodic activity timed near the peak in the temporal factor (Fig. 3c, center). Any given neuron tended to display high neuron factor values in multiple components (Fig. 3c, left), thus reflecting the occurrence of multiple activity episodes for any one neuron (Fig. 2a). The co-activation of neurons within a given component (i.e., multiple neurons with a high neuron factor; Fig. 3c, left) revealed the temporal overlap between episodic activity from different neurons (Fig. 2b). Further, the

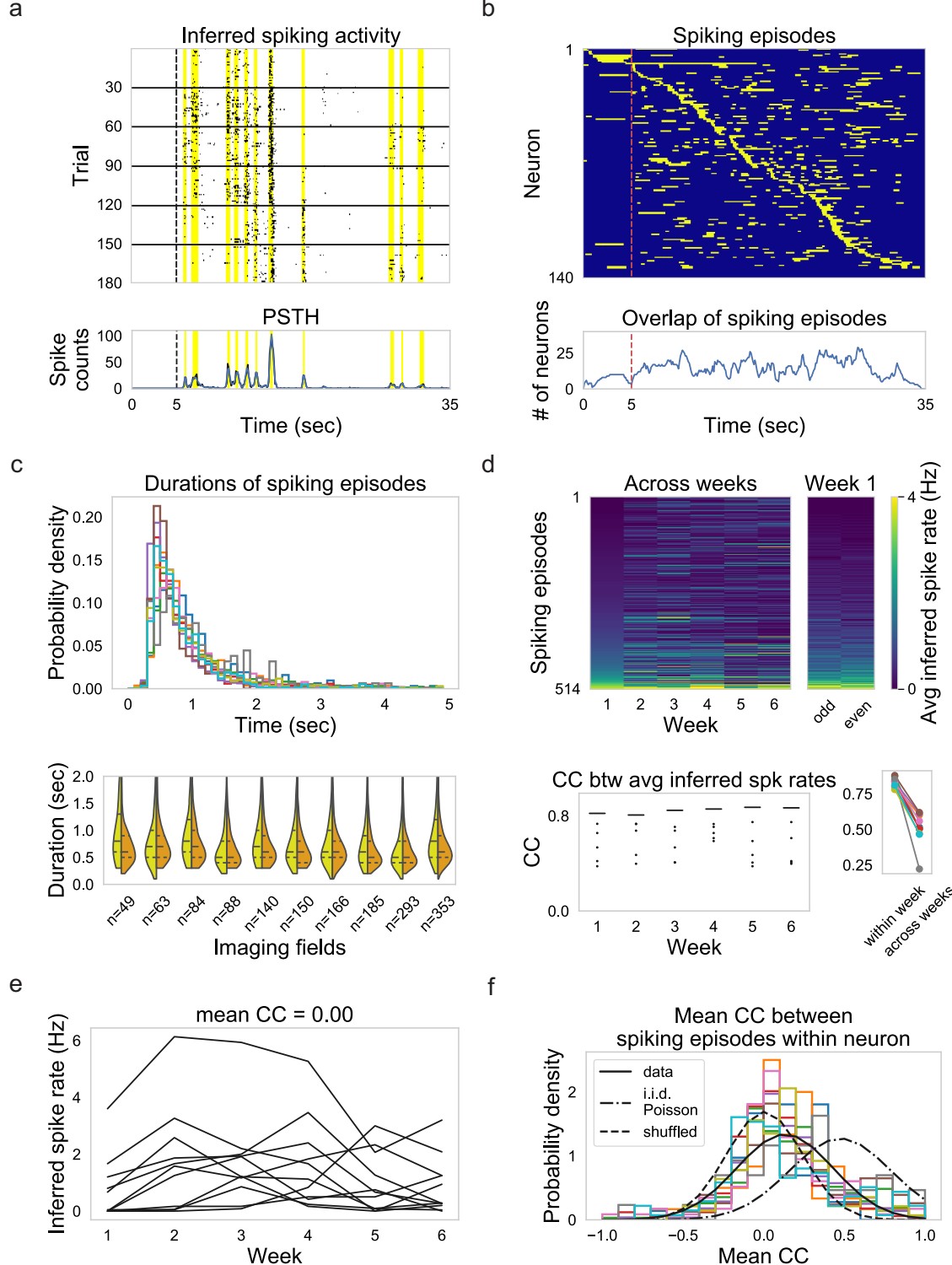

diverse variation of the trial factor values (Fig. 3c, right) reflected the diverse gain variability of episodic activity (Fig. 2d), even for any given neuron (Fig. 2e, f).

In summary, the TCA dimensionality reduction confirmed in an unsupervised manner the episodic activity of single neurons (Fig. 2a), the temporal overlap of episodic activity from different neurons (Fig. 2b), and the diversity of week-to-week fluctuations of episodic activity within a given neuron (Fig. 2d–f). In conclusion, the results from the TCA analysis (Fig. 3c) support the hypothesis that cortical coordination resides at the level of

episodic activity, rather than at the level of neurons, as is commonly assumed[36].

Visual inspection of the trial factors across weeks indicated vastly diverse dynamics across weeks for different components. To illustrate this diversity of dynamics, we sorted the components by their trial factors using K-means clustering, choosing 5 or 6 clusters (Fig. 3c and Supplementary Fig. 3). Within each thus determined cluster, we further ordered the components by the time to peak in their temporal factors. This reorganization of the TCA analysis display revealed two important insights. First, trial

**Fig. 2 Single neuron responses consist of episodic activity with distinct episode-specific rate variations across weeks. a** Top: inferred spiking activity of the same neuron shown in Fig. 1b. Bottom: peristimulus time histogram (PSTH) (black) and smoothed PSTH (blue) of the same neuron. Shaded areas (yellow) denote spiking episodes for this neuron. **b** Top: spiking episodes for all the neurons in the example imaging field. Neurons are ordered by the latency of their spiking episodes with the highest spiking rates. Bottom: number of neurons with overlapped spiking episodes. **c** Top: distributions of durations of spiking episodes from all imaging fields. Different colors denote different imaging fields. Bottom: distribution of durations of spiking episodes defined from PSTH of trials across weeks (yellow) plotted against the distribution of durations from PSTH of trials within weeks (orange). **d** Top: averaged inferred spike rates over trials of all the spiking episodes in one example imaging field are plotted for different weeks and for even and odd trials in week 1. spiking episodes are ordered by their averaged inferred spike rates during week 1. Bottom left: correlation coefficients (CC) between averaged inferred spike rates of week pairs (dots) and even/odd trials within the week (lines) are shown for the example imaging field across weeks. Bottom right: CC within each week averaged across weeks is plotted against CC across weeks averaged across all the week pairs (one-sided Wilcoxon signed-rank test, $p = 0.0025$, ten imaging fields). Different colors denote different imaging fields. Colormap maximum value is set to 4 Hz. **e** Mean inferred spike rate during each spiking episode in the example neuron varies across weeks. **f** Histogram of mean CC between mean inferred spike rates during spiking episodes within the same neuron. Different colors denote different imaging fields. The black solid line is a Gaussian curve fitted to the distribution of mean CC from all the imaging fields (mean 0.13, s.t.d. 0.30). The black dash-dotted line is a Gaussian curve fitted to the distribution of mean CC between simulated independent and identically distributed Poisson spike trains with the firing rates of a randomly selected spiking episode for a given neuron (mean 0.46, s.t.d. 0.31). The black dashed line indicates the chance level, which is a Gaussian curve fitted to the distribution of mean CC between spiking episodes with independently shuffled weeks (mean 0.0036, s.t.d. 0.23). Only neurons with more than one spiking episode were included in this analysis.

factors changed in a distinctly different manner across weeks for different clusters of components. For instance, while the trial factors for the first cluster of components were largely homogeneous across weeks, the trial factors for the second cluster largely faded away after the second week. Of functional significance, with such vanishing trial factors, the second cluster of components would contribute little to a stimulus representation in week 4 and beyond. We observed such diverse dynamics of trial factors across weeks for all imaging fields studied (Fig. 3d, e and Supplementary Fig. 3). Second, within each cluster of components, the pronounced peaks in the temporal factors were largely evenly distributed across the duration of the trial. Assuming that the peaks in the temporal factors (or equivalently the spiking episodes; see Fig. 2) contribute to cortical stimulus representation, the even distribution of these peaks suggests that every moment in the movie was evenly represented, however by different groups of neurons at different weeks. In conclusion, the diverse dynamics of trial factors across weeks for different components indicates a fluid long-term stimulus representation in visual cortex. Importantly, the fluid stimulus representation was structured at the level of episodic activity rather than the neuron.

**Stable manifolds exist in unstable population activity**. As expected from the interconnected nature of cortical circuits[37], we observed population-wide correlated neural fluctuations summarized by TCA components in the previous section (Fig. 3). Does a stable representation emerge from unstable population responses? To answer this question, we searched for a stable neural manifold using dimensionality reduction.

We mapped the high-dimensional denoised neuronal population responses (reconstructed responses; Fig. 3b) of episodic activity onto a low-dimensional space (manifold) and investigated the stability of the activity on this manifold (Fig. 4). For N recorded neurons, the denoised instantaneous population response $\Delta F/F$ is a point in an $N$-dimensional state space. In an attempt to preserve the manifold topology of neuronal population responses (Fig. 2a, b), we chose a mapping such that nearby points in the high-dimensional state space would also be adjacent in the resulting low-dimensional space. Since the structure of the presumed intrinsic manifold was not known a priori, we adopted the unsupervised algorithmic approach, Isomap, for the mapping (see "Methods"; ref. [38]).

For visualization purposes, we plotted the mapped population responses in the first 3 Isomap dimensions (i.e., three eigenvectors with the largest eigenvalues of the geodesic distance matrix;

Fig. 4a). Each dot is a nonlinear projection of the instantaneous population activity into this three-dimensional space. Interestingly, most of the dots resided on a ring-shaped low-dimensional manifold, forming well-aligned trajectories of neural activity across trials (Fig. 4a and Supplementary Fig. 4a). Note that the ring structure of the manifold arose from the looped trial structure of the visual stimulus. If the stimulus were repeated but not looped in time, i.e., interleaved with different stimuli between trials, we would expect to see a line structure for the manifold.

To quantify the stability of these trajectories across trials, we projected all trajectories against a given Isomap dimension and compared projected trajectories across all trials (Fig. 4b). From the visual inspection of the projected trajectories in the first three Isomap dimensions, we obtained a sense of the stability of these trajectories across trials and sessions. For further quantification, we used the average correlation coefficient of these projected trajectories from all pairs of projected trajectories as a measure of stability across trials (Fig. 4c). Stability was high for the first few Isomap dimensions but beyond those decreased with increasing Isomap dimension.

In conclusion, this unsupervised analysis showed that stable low-dimensional latent variables exist in population activity consisting of unstable single neuronal responses (Fig. 1) that are sparse and temporally structured into episodic activity (Figs. 2, 3). This finding is likely to be of functional significance. Even though the high-dimensional population vector contains considerable variability, there exists a stable low-dimensional subspace for potentially stable representation of visual stimuli. The discovery of a stable manifold set the stage for stable stimulus representation.

**The manifold mediates a stable representation of the time within the movie clip**. To extract the stimulus representation potentially encoded in the stable neural manifold, we applied spline parameterization for unsupervised decoding (SPUD)[39] to population activity embedded in the first few Isomap dimensions. Here we only showed results for the first two Isomap dimensions for visualization, but the following results also hold for up to the first five dimensions (Supplementary Fig. 5a). The decoding process consisted of the following steps (Fig. 5a). First, we randomly split the two-dimensional neural manifold into a training set (80%) and a test set (20%). Second, we fit a one-dimensional spline to the training set, and then assigned coordinates to the fitted spline. Third, we assigned each dot in the test set a value according to the coordinate of its nearest point on the spline. Last, we circularly shifted or flipped the coordinates on the spline such that we achieved the best decoding performance (circular least

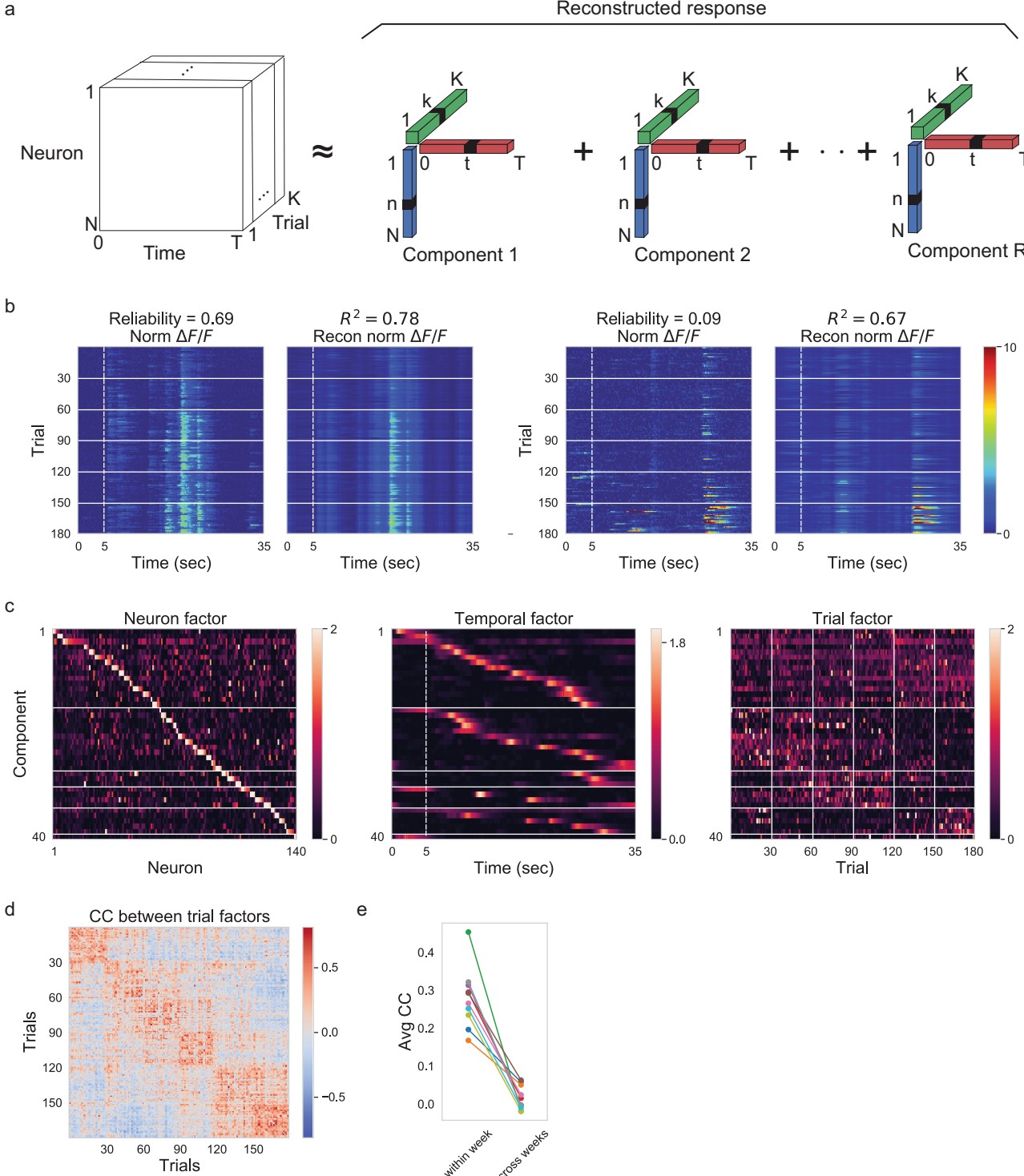

mean squared error) for time in the movie of the test set. We did this because when we assigned coordinates to the spline, the origin and direction of the coordinates were arbitrarily determined. To match the assigned coordinates with the actual time, we need to determine the origin and direction of coordinates using the test set. The decoded time α closely traced the actual time $t$ in the movie for population activity across weeks (Fig. 5b). We summarized the decoding error (circular absolute difference between $t$ and $\alpha$) from all the recorded imaging fields (Fig. 5c). As a comparison, the decoding error of SPUD was significantly lower than the decoding error from that of linear decoders (Supplementary

Fig. 5b). In general, the decoding performance improved with an increasing number of recorded neurons in the imaging field (Fig. 5c). To further investigate the stability of neural representation of time in the trial, we also trained SPUD on neural data from odd trials in week 1 and tested its performance on neural data from even trials in week 1 and trials from other weeks. The decoding errors pooled from later weeks were not significantly different from the decoding errors for week 1 across imaging fields ($p = 0.11$, two-sided Mann−Whitney U test; Supplementary Fig. 5c, d). This additional analysis showed that week-to-week variability does not affect the coding of time in the movie.

**Fig. 3 Latent factors resembling episodic activity with gain changes capture the across-week fluctuations. a** Schematic of tensor component analysis (TCA). Neural activity ($\Delta F/F$) is organized into a third-order tensor with dimensions $N \times T \times K$. TCA approximates the data as a sum of outer products of three vectors from $R$ components: neuron factors describe the weights of each neuron to that component, temporal factors describe the temporal dynamics of each component, and trial factors describe the modulation of the component across trials. **b** Normalized $\Delta F/F$ responses and reconstructed $\Delta F/F$ from 40 TCA components of two example neurons from the example imaging field. Reliability was defined as averaged correlation-coefficient between pairs of single-trial responses[3]. **c** Neuron, temporal, and trial factors of nonnegative TCA with 40 components for the example imaging field. Colormap maximum values are set to 2 for neuron factors and trial factors. We ordered components according to the K-means clustering on their trial factors. Within each thus determined cluster, we further ordered the components by the time to peak in their temporal factors. We ordered neurons in the neuron factors by their dominant components. **d** Correlation coefficient (CC) between trial factors shown in (**c**). **e** CC between trial factors averaged across trial pairs within week plotted against CC between trial factors averaged across trial pairs across weeks. Different colors denote different imaging fields. The week-to-week variability of trial factors was significantly larger than the corresponding trial-to-trial variability within each week (for avg CC, one-sided Wilcoxon signed-rank test, $p = 0.0025$, ten imaging fields).

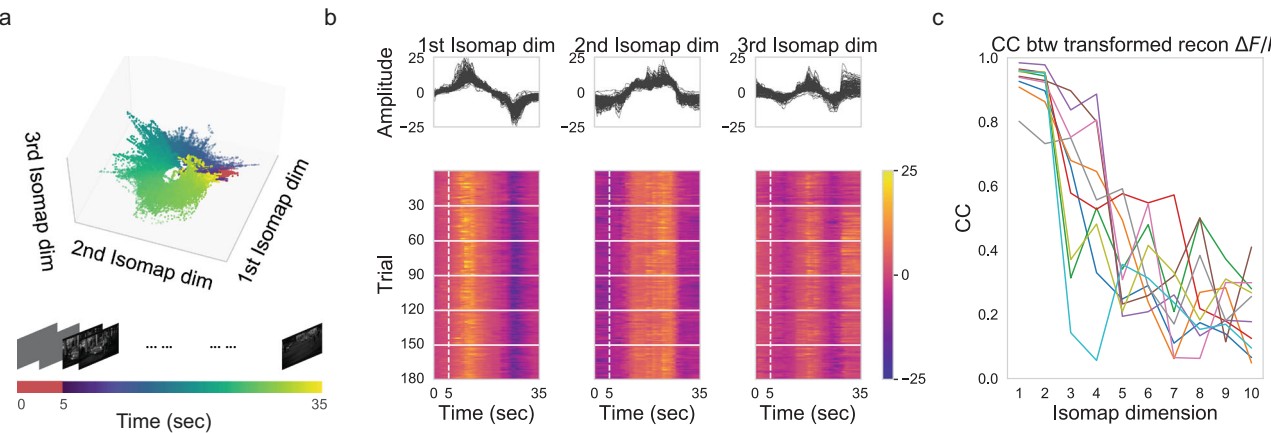

**Fig. 4 Stable manifolds exist in unstable population activity. a** Three-dimensional neural trajectories extracted from reconstructed (denoised) $\Delta F/F$ populational activity across weeks from the example imaging field using Isomap. Each dot represents instantaneous population activity. The color of the dot indicates the corresponding time in the trial. **b** Upper panels: neural trajectories along the first three Isomap dimensions were plotted separately. Lower panels: the same data as shown in the upper panels were displayed as heatmaps to show the trial-to-trial variability clearly. **c** Correlation coefficients (CC) between transformed reconstructed $\Delta F/F$ (neural trajectories) across trials along each Isomap dimension are plotted for all ten imaging fields. Different colors denote different imaging fields.

One of the key reasons for the high decoding performance of population activity as analyzed by SPUD resides in the isometric representation[39]. Time in the movie was evenly represented along the fitted spline direction. In other words, equal amounts of population activity variations along the spline direction contributed to equal amounts of change across time in the trial. This isometric representation was related to the evenly distributed episodic activity across time in the trial, as shown by TCA components (Fig. 3c). If we removed episodic activity during a certain time window in the trial from the population activity, then the corresponding section in the manifold ring would collapse into the hyperplane perpendicular to the spline direction (Supplementary Fig 6).

Due to the high trial-to-trial variability in population activity, the ring-shaped neural manifold had many outlier dots. The outlier dots in the center of the ring corresponded to low amplitude of population activity, while outlier dots on the outside of the ring corresponded to high amplitude of population activity (Supplementary Fig. 4b). The decoder failed at a few outlier dots. However, most of the neural variability seemed to be perpendicular to the direction of the fitted spline, thus, harmless to decoding. This observation gave us a hint about the mechanism that maintains stable neural correlates in the face of dynamical population activity.

**Both week-to-week fluctuation and trial-to-trial variation within the week is restricted to non-coding directions**. In order to quantify to what extent neural variability influences the

stimulus coding, we calculated the variance of instantaneous population activity on the manifold along the direction parallel or perpendicular to the fitted spline. Specifically, we computed the parallel and perpendicular component of the instantaneous population activity variance employing the following steps. First, we reconstructed $\Delta F/F$ population activity based on 40 TCA components (Fig. 6a). Second, we used Isomap to project the population activity of all trials into a two-dimensional space. Third, as illustrated in Fig. 5, we separated the projected instantaneous population activity into a training set and a test set. Fourth, we calculated the fitted spline to the training set. Fifth, we computed the coordinates on the spline, based on the test set data (Fig. 6b, left panel). Sixth, for each time point in the movie, we calculated the variance of instantaneous population activity in the test set along the direction parallel or perpendicular to the spline (Fig. 6b, right panel). Finally, we summarized the variance for all the time points in the movie (Fig. 6c). The variance of population activity along the spline direction was significantly smaller than that perpendicular to the spline direction. This observation held for eight out of ten imaging fields (Fig. 6d). In this computational framework, the spline direction signifies the stimulus coding direction. In conclusion, the comparatively small contribution of neural variability to stimulus encoding direction directly explains why the high neural variability we observed in spiking episodes (Fig. 2) did not harm the decoding performance of SPUD (Fig. 5).

The neural variability we measured here consisted of two portions: week-to-week fluctuations and trial-to-trial variability within each week. Are they both restricted to the non-coding

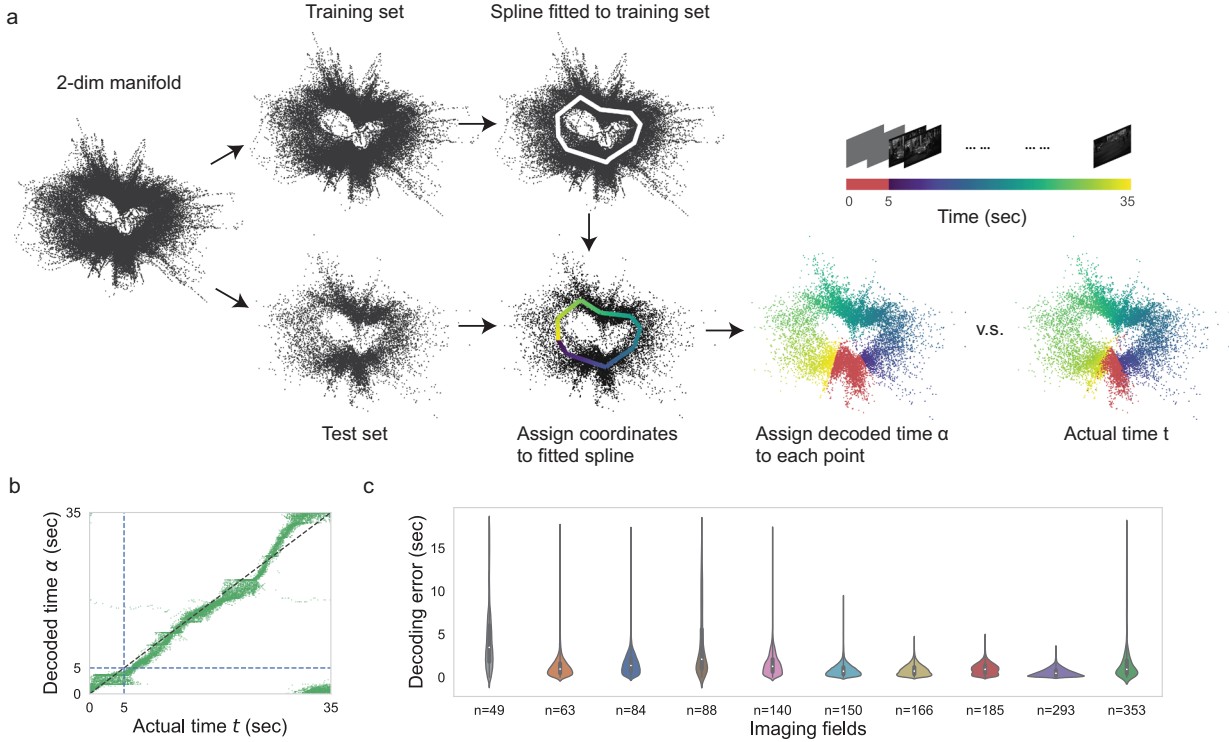

**Fig. 5 The manifold mediates a stable representation of the time within the movie clip. a** Illustration of the unsupervised method with data from the example imaging field ($n = 140$): first, we projected reconstructed DF/F responses into the first two Isomap dimensions, each dot denotes instantaneous population activity; second, we randomly pick 80% of the instantaneous population activity as training set and rest of them as test set; third, we fitted a spline to the neural manifold of the training set and assigned coordinates with randomly picked origin to the fitted spline; finally, we shifted and flipped the coordinates on the fitted spline to match with the actual time and assigned decoded time to each point in the test set by its nearest coordinate on the spline. **b** Decoded time from the neural manifold plotted against actual time in the movie for the example imaging field ($n = 140$, 12,600 timepoints). **c** Violin plots showed decoding error (absolute circular difference between decoded time and actual time) for all the imaging fields (14,700 timepoints for imaging field $n = 63, 84, 185, 293$, and 353; 12,600 timepoints for imaging field $n = 140, 166$; 10,500 timepoints for imaging field $n = 49, 88$, and 150). Inner boxplots show median: center white point; interquartile range: box; and data range (maximum 1.5 interquartile range): whiskers. Imaging fields were ordered by the number of recorded neurons.

direction? To answer this question, we quantified week-to-week variability and trial-to-trial variability within each week separately. For week-to-week fluctuations, first, we calculated the trial-averaged projected population activity in the two-dimensional space for each week (Fig. 6e). Second, we calculated the variance of those trial-averaged instantaneous population activity across weeks along the direction parallel or perpendicular to the spline. Finally, we summarized the variance for all the time points in the movie (Fig. 6f, left panel). The significantly larger week-to-week variance along the direction perpendicular to the spline compared with that of parallel direction suggested that the week-to-week fluctuation was also constrained to the non-coding direction. For trial-to-trial variability within each week, first, we calculated the variance of single-trial population activity for each week separately. Second, we summarized the variance for all the weeks and all the time points in the movie (Fig. 6f, right panel). The trial-to-trial variability within each week was larger along the direction perpendicular to the spline compared with that of parallel direction. Furthermore, the same observation held for most imaging fields (Fig. 6g). In conclusion, both week-to-week fluctuations and trial-to-trial variability within each week were restricted to the non-coding direction.

**The precisely timed episodic activity constrains neural variability to non-coding directions.** How is neural variability largely constrained to the direction perpendicular to stimulus coding direction? Is it caused by the reproducible timing of episodic

activity, by the coordination between different episodes, or by the combination thereof? To answer these questions, we applied the previous analyses to shuffled reconstructed $\Delta F/F$ population activity.

First, we checked whether the neural manifold was an artifact of the method by applying Isomap to shuffled data. To remove both the reproducible timing of episodic activity and the coordination of episodic activity across neurons in the shuffled data, we circularly time-shifted reconstructed $\Delta F/F$ responses by a random amount for every trial of each neuron independently (Fig. 7a). In other words, only the temporal statistics of $\Delta F/F$ responses were kept. As expected, neural trajectories from different trials were not aligned (Fig. 7b). However, trajectories were continuous instead of being a noisy point cloud. Such continuous trajectories arise from the smooth nature of shuffled reconstructed $\Delta F/F$ responses. This sanity check showed that the ring structure of the neural manifold (Fig. 6b) arose from the timing and coordination of the population activity and was not an artifact of the method.

Second, we checked whether the reproducible timing of episodic activity was sufficient to constrain the neural variability by applying Isomap to shuffled data with preserved trial structure. To merely remove the coordination between different episodes, but to maintain the amplitude of the covariance of neural activity, we chose to shuffle TCA factors instead of shuffling reconstructed population activity. In contrast, shuffling reconstructed population activity would decrease the covariance between neural

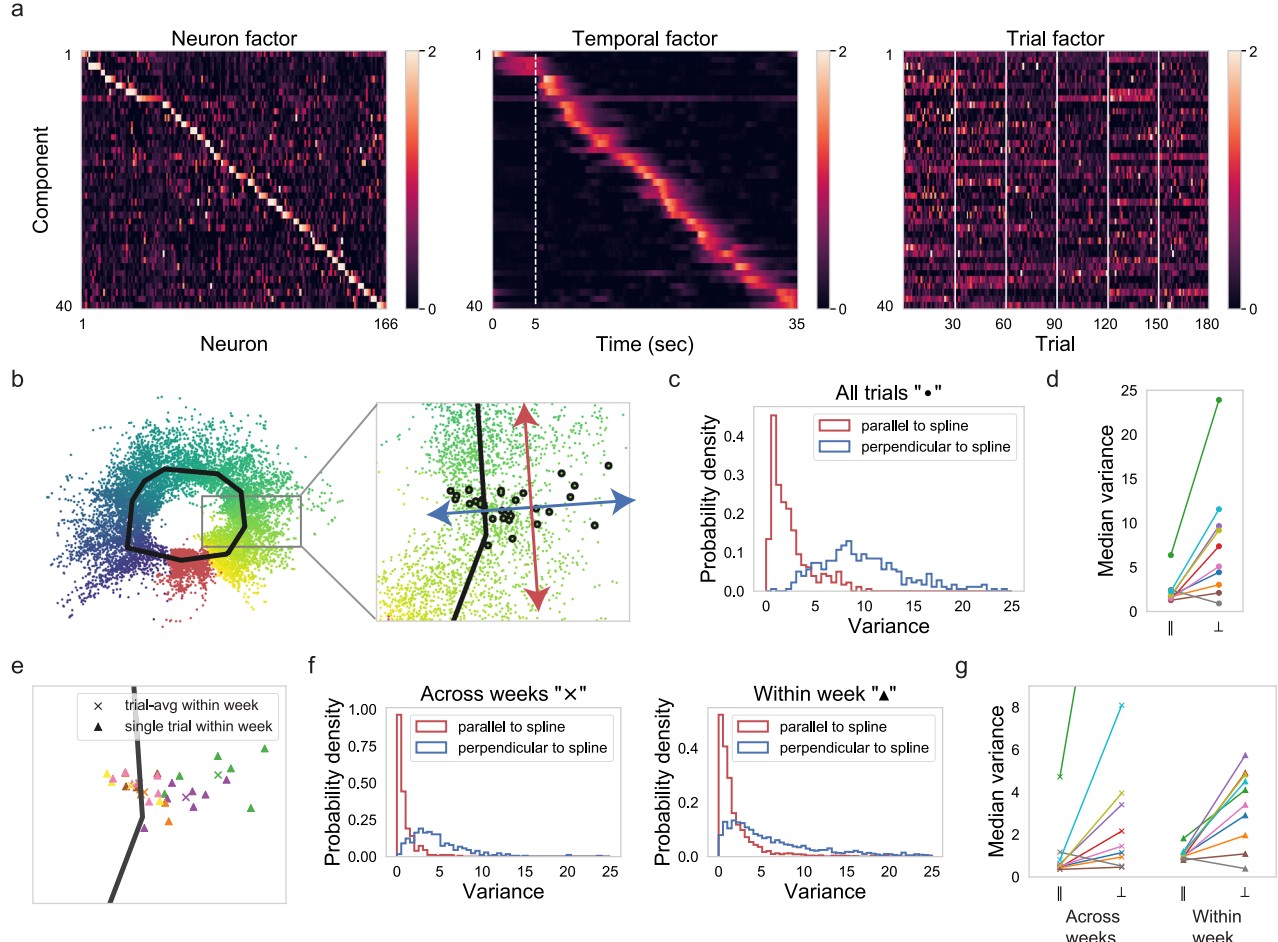

**Fig. 6 Both week-to-week fluctuation and trial-to-trial variation within the week is restricted to non-coding directions. a** TCA components of one imaging field ($n = 166$). We ordered components by the time to peak in their temporal factors. We ordered neurons in the neuron factors by their dominant components. Colormap maximum values are set to 2 for all the factors. **b** Left: two-dimensional neural manifold extracted from reconstructed (denoised) $\Delta F/F$ population activity ($n = 166$) across weeks using Isomap. Each dot represents instantaneous population activity in the test set. The color of the dot (same colormap as Fig. 4a) indicates the corresponding time in the trial. The black line is the fitted spline to the training set. Right: zoom-in view on the neural manifold. Instantaneous population activity corresponding to 28 s in the trial was highlighted with black shade. The Blue arrow denotes the direction perpendicular to the spline, and the red arrow denotes the direction parallel to the spline. **c** Histogram of the variance of population activity parallel or perpendicular to the spline for the imaging field shown in (**a**, **b**). **d** Median variance of population activity parallel to the spline plotted against median variance of population activity perpendicular to the spline for all the imaging fields. The variance of population activity parallel to the spline was significantly smaller than the variance perpendicular to the spline for most imaging fields (one-sided Wilcoxon signed-rank test, 350 timepoints, $p$-values for different imaging fields are $3.93 \times 10^{-31}$ ($n = 63$), $2.61 \times 10^{-55}$ ($n = 84$), $1.78 \times 10^{-40}$ ($n = 140$), $2.61 \times 10^{-58}$ ($n = 150$), $2.91 \times 10^{-59}$ ($n = 166$), $7.37 \times 10^{-57}$ ($n = 185$), $2.06 \times 10^{-59}$ ($n = 293$), $3.03 \times 10^{-31}$ ($n = 353$)), except for two imaging fields ($p = 1.0$ (gray, $n = 49$) and $p = 0.997$ (brown, $n = 88$)). The two outliers are from two imaging fields whose neural manifold didn't have a clear ring shape (Supplementary Fig. 5a). **e** The same zoom-in view on the neural manifold as shown in (**b**) (right). Each triangle represents instantaneous population activity within a week. The color of the triangle denotes different weeks. Each cross represents the trial-averaged instantaneous population activity within a week. The color of the cross also denotes different weeks. **f** Left: histogram of the variance of trial-averaged population activity within a week parallel or perpendicular to the spline for the imaging field. Right: histogram of the variance of single-trial population activity within a week parallel or perpendicular to the spline for the imaging field. **g** Median variance of trial-averaged population activity or single-trial population within a week parallel to the spline plotted against median variance of population activity perpendicular to the spline for all the imaging fields. Y-axis is clipped at 9 for visualization. The variance of population activity parallel to the spline was significantly smaller than the variance perpendicular to the spline for both across weeks and within a week cases for most imaging fields (across weeks: one-sided Wilcoxon signed-rank test, 350 timepoints, $p$-values for different imaging fields are $1.45 \times 10^{-27}$ ($n = 63$), $1.41 \times 10^{-39}$ ($n = 84$), $1.55 \times 10^{-33}$ ($n = 140$), $5.13 \times 10^{-57}$ ($n = 150$), $8.50 \times 10^{-58}$ ($n = 166$), $1.29 \times 10^{-54}$ ($n = 185$), $1.26 \times 10^{-57}$ ($n = 293$), $1.25 \times 10^{-29}$ ($n = 353$), two outliers $p = 1.0$ (gray, $n = 49$), $p = 0.99$ (brown, $n = 88$); within a week: one-sided Wilcoxon signed-rank test, number of samples = 350 timepoints × number of weeks, $p$-values for different imaging fields are $7.36 \times 10^{-97}$ ($n = 63$), $1.04 \times 10^{-219}$ ($n = 84$), $1.53 \times 10^{-177}$ ($n = 140$), $6.13 \times 10^{-171}$ ($n = 150$), $6.23 \times 10^{-273}$ ($n = 166$), $0.0$ ($n = 185$), $0.0$ ($n = 293$), $2.99 \times 10^{-120}$ ($n = 353$), two outliers $p = 1.0$ (gray, $n = 49$), $p = 1.0$ (brown, $n = 88$)).

activity across neurons, in addition to removing the coordination between episodic activity. For each TCA component, we randomly shuffled the neuron order in the neuron factor, and we circularly shifted the temporal factor and the trial factor by a random amount (Fig. 7c). Thus, by shuffling the factors for each

component independently, we removed all the significant coordination between episodic activity. As expected, the removal of coordination between episodic activity resulted in a new manifold and a new spline (Fig. 7d). However, the variability of reconstructed population activity (based on shuffled TCA factors)

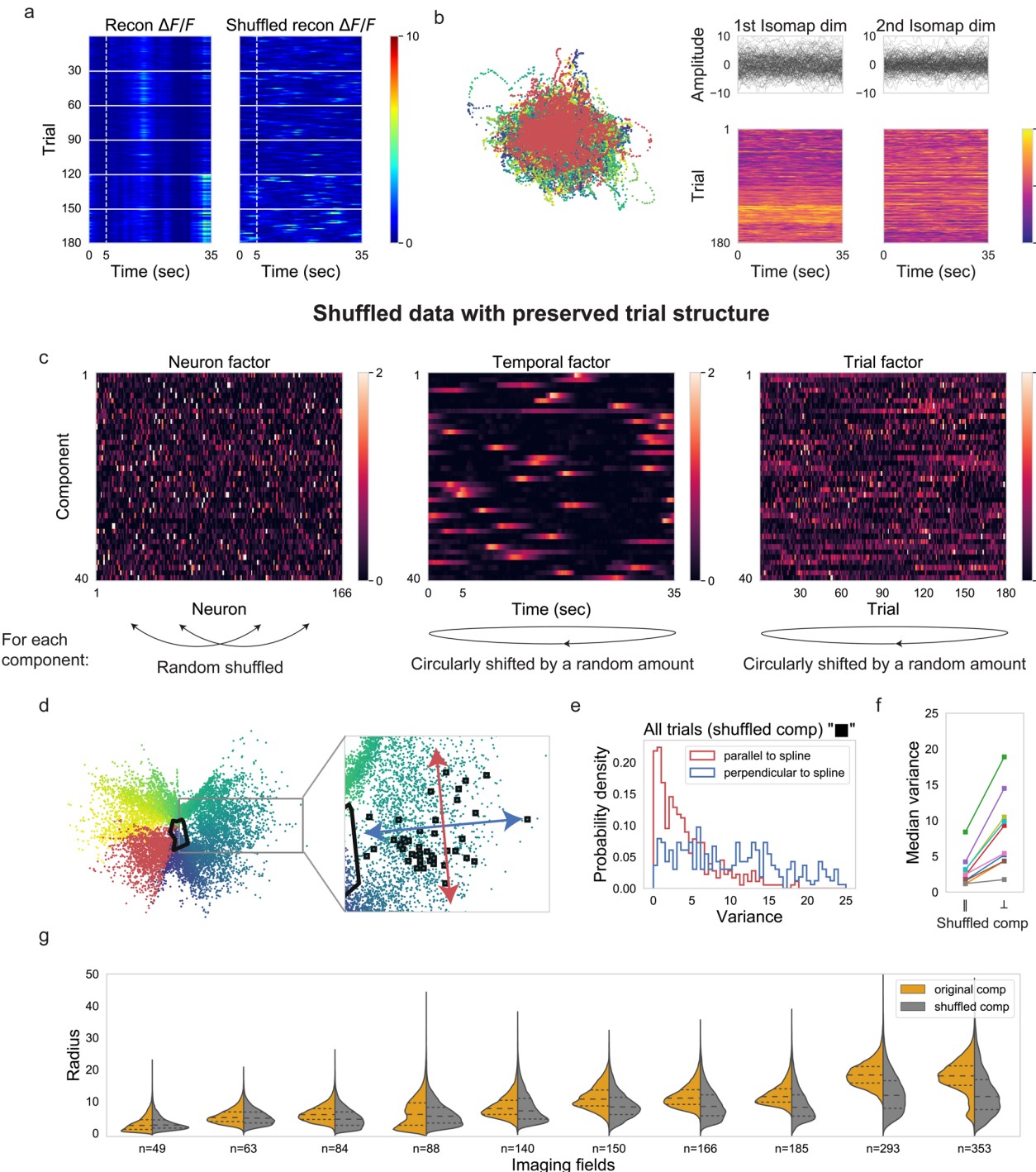

continued to be largely constrained to the direction perpendicular to the spline (Fig. 7d). The smaller variability of population activity parallel to the spline is visible in the separation of dots of different colors, where color indicates the time in the trial of the instantaneous population activity (Fig. 7d). Indeed, the quantification of variance showed that the amplitude of neural variability along the spline was significantly smaller than that perpendicular to the spline (Fig. 7e). Moreover, the significant difference between variance along the direction parallel and perpendicular to the spline held for shuffled data with preserved trial structure from all the imaging fields (Fig. 7f). In conclusion, the fact that episodic activity is precise in time across trials

(Fig. 2c) alone is sufficient for constraining neural variability to the direction perpendicular to the stimulus encoding direction. In contrast, the coordination among episodic activity plays no role in this constraint.

However, coordination between episodic activity is essential for uniquely representing time points in the trial. The neural manifold of shuffled data with preserved trial structure had a collapsed ring structure (Fig. 7d and Supplementary Fig. 7a) in contrast to the clear ring structure from original data (Figs. 5a, 6b). The collapsed ring structure would lead to ambiguous decoding due to the overlap between instantaneous population activity from different time points in the trial (Supplementary Fig. 7b).

**Fig. 7 The precisely timed episodic activity constrains neural variability to non-coding directions. a** Reconstructed $\Delta F/F$ responses and shuffled reconstructed $\Delta F/F$ responses of two example neurons from the imaging field ($n = 166$). The single neuron response was circularly shifted by a random amount independently for each trial for shuffling. **b** Left: two-dimensional neural manifold extracted from shuffled reconstructed $\Delta F/F$ population activity (example single neuronal shuffled responses shown in (**a**)). The same colormap was used as in Fig. 6b, left panel. Right: neural trajectories along the first two Isomap dimensions (the same as shown in the left panel) organized in trial by time matrices. Here we set the number of nearest neighbors of ISOMAP to be 100 (see "Methods"). **c** TCA components of the imaging field ($n = 166$) with shuffled factors. For each component, we independently shuffled neuron order in the neuron factor, circularly shifted the time factor and the trial factor by a random amount. Components with shuffled factors were ordered again in the same fashion as Fig.6a. **d** Two-dimensional neural manifold extracted from reconstructed (denoised) $\Delta F/F$ population activity ($n = 166$) from components with shuffled factors using Isomap. Each dot represents instantaneous population activity in the test set. The black line is the fitted spline to the training set. Instantaneous population activity corresponding to 16 s in the trial was highlighted with black shade. **e** Histogram of the variance of neural variability parallel or perpendicular to the spline for reconstructed (denoised) $\Delta F/F$ populational activity ($n = 166$) from components with shuffled factors (i.e., shuffled data with preserved trial structure). **f** Median variance of neural variability parallel to the spline plotted against median variance of neural variability perpendicular to the spline for all the imaging fields for reconstructed activity from components with shuffled factors. The variance of population activity parallel to the spline was significantly smaller than the variance perpendicular to the spline (one-sided Wilcoxon signed-rank test, 350 timepoints, $p$-values for different imaging fields are $1.34 \times 10^{-14}$ ($n = 49$), $6.65 \times 10^{-58}$ ($n = 63$), $2.24 \times 10^{-43}$ ($n = 84$), $4.93 \times 10^{-25}$ ($n = 88$), $3.76 \times 10^{-24}$ ($n = 140$), $2.05 \times 10^{-33}$ ($n = 150$), $5.16 \times 10^{-36}$ ($n = 166$), $9.39 \times 10^{-40}$ ($n = 185$), $7.87 \times 10^{-46}$ ($n = 293$), $1.86 \times 10^{-13}$ ($n = 353$)). **g** Radius (distance to the center of the point cloud) distribution of points on the neural manifold from original TCA components plotted against radius distribution from TCA components with shuffled factors for all the imaging fields. Except for one imaging field ($n = 49$), the radius of points on the neural manifold from original TCA components was significantly larger than the radius from TCA components with shuffled factors (one-sided Mann−Whitney U test, 73,500 timepoints for imaging field $n = 63, 84, 185, 293,$ and $353$; 63,000 timepoints for imaging field $n = 140, 166$; 52,500 timepoints for imaging field $n = 49, 88,$ and $150$, $p$-values for different imaging fields are $1.0$ ($n = 49$, outlier), $6.68 \times 10^{-113}$ ($n = 63$), $0.0$ ($n = 84$), $1.06 \times 10^{-5}$ ($n = 88$), $2.40 \times 10^{-248}$ ($n = 140$), $0.0$ ($n = 150, n = 166, n = 185, n = 293, n = 353$)).

We quantified the shape of the neural manifold for original and shuffled data with preserved trial structure by calculating the distance from each dot representing instantaneous population activity to the center of the manifold (see "Methods"). For nine out of ten imaging fields, the neural manifold of shuffled data with preserved trial structure had a more collapsed ring structure than the manifold of original data, as shown by a significantly smaller radius (Fig. 7g). The collapsed ring structure of shuffled data showed that stable representation of the time in the natural movie requires not only that some neurons display reliable responses over sessions (shuffled data also have neurons with reliable responses), but also coordination between episodic activity.

In summary, both the nature of the precise episodic activity and the coordination between different activity episodes contributes to encode time in the natural movie. However, episodic activity reproducible in time alone is sufficient for restricting neural variability to non-coding directions.

## Discussion

We showed that single neuronal responses to the natural movie in V1 consisted of episodic activity with variability in gain across weeks. Importantly, we found a stable low-dimensional subspace inside the highly variable high-dimensional neural space. Time in the movie was represented on a one-dimensional ring manifold isometrically, where equivalent changes on the ring indicated equivalent changes in time. Moreover, we found that the limited influence of neural variability and week-to-week fluctuations on the stable representation of the natural movie was mediated by the fact that most of the neural variability was constrained in the non-coding direction, augmenting the previous literature on population coding and neural variability[19,40–42]. Furthermore, we found that stable episodic activity was sufficient for restricting neural variability to non-coding directions independent of coordination between episodic activity.

To study the neural representation in V1, it is common practice in the field to measure tuning curves (trial-averaged single-neuron activity) with respect to external variables[43–45] or decode external variables from neural activity with supervised methods, such as linear decoders[11,46,47]. In contrast, recent work introduced unsupervised methods in revealing the internal representation using neural data alone without reference to external variables[39,48]. Here, we identified an internal representation of time in the natural movie by parameterizing the neural manifold, without using any external information or prior assumptions.

There are several advantages in the dissociation of internal and external variables. First, such dissociation avoids the biases introduced by the chosen external variables. One caveat of interpreting the neural activity through the lens of the chosen external variable is that the encoded variable might be different but correlated with the chosen external variable. Thus, non-trivial tuning curves or supervised decoding results do not necessarily reveal the actual neural representation. Second, dissociation of internal and external variables permits discovering representation of cognitive variables. It is possible that the internal variable represents the animal's inference about an external variable. For example, as hypothesized by the sampling-based neural variability theory[49,50], neural variability in V1 might represent the perceptual uncertainty of certain visual features. In the future, it will be interesting to investigate whether the thickness of the ring manifold (Fig. 6) reflects the animal's perceptual uncertainty of certain scenes in the movie.

Even though population activity may never visit the same state in the high dimensional space, there exists a stable readout direction as indicated by the fitted spline (Fig. 5). The liquid state machine (LSM)[51], a computational paradigm for recurrent neural networks, describes a similar situation. Instead of viewing neural networks as "feature detectors", LSM views the network as liquid, continuously receiving external perturbations. Although the liquid neural trajectory keeps changing across time, we can get a stable readout by training a linear readout unit. Note that our work is different from LSM in the readout method, as we obtained stable readout in an unsupervised manner. LSM suggests that trial-to-trial variability reflects an accumulation of information instead of noise, as recurrent network activity implicitly contains the previous external perturbations. This recurrent-network perspective can be instructive for our future work. In our work, we found that trial-to-trial variability is mostly constrained in the direction perpendicular to the spline direction (Fig. 6). However, we did not interpret the latent variables encoded in other directions except for the spline direction. Moreover, recent

works suggest that V1 encodes various behavior and state variables besides visual-related variables[3,52,53]. A new experimental design with behavior or state recordings might provide a more complete picture of internal representation in V1.

The low-dimensional internal representation offered us a better reference point to understand neural dynamics than the high-dimensional population activity[54]. As a promising future direction, it would be informative to study neural dynamics on or off the manifold with perturbations[55]. One way of perturbation is to modulate the visual stimulus[56,57]. For example, on some of the trials, we propose to overlay flash dots with some frames in the natural movie[58] and observe whether the neural trajectories first deviate from the ring manifold and then flow back. Another way of perturbation is to directly control neural activity with optogenetics[59–61]. As suggested by the TCA analysis, episodic activity shared across neurons was the building block for the ring manifold (Fig. 3). It will be interesting to see how the optogenetically mediated changes of spiking timing or amplitude of episodic activity impact population dynamics on or off the manifold.

Previous studies[52,62,63] suggest behavioral variables such as locomotion and arousal could lead to gain modulations in single neural activity. However, the reported modulations by behavioral variables were homogeneous across neurons[63] and whether changes in behavioral variables would contribute to the heterogeneous gain modulations across episodic activity within the same neuron (Fig. 2e, f) is not clear. Furthermore, although we checked the limited impact of eye movement and pupil size on the stability of neuronal responses in an earlier work[25], how behavioral variables would affect the neural trajectories remains to be explored in the future.

At the neural circuit level, there are several possible mechanisms that could lead to the observed drift. First, the turnover of boutons and dendritic spines in V1 at the baseline condition[64,65] would cause changes in the recurrent inputs to single neurons. Second, potential changes in the feedforward connectivity from LGN to V1 or drift in LGN responses would cause drift in the feedforward inputs to single neurons. Third, slowly varying top-down inputs related to visual information processing could contribute to the drift as well. Model investigations and simultaneous chronic recordings from LGN or high-order visual areas would be helpful for distinguishing contributions from these potential mechanisms in the future.

## Methods

**Animals**. For imaging visual cortical responses, a Emx1-Cre (Jax Stock #005628) x ROSA-LNL-tTA (Jax Stock #011008) x TITL-GCaMP6s (Jax Stock #024104) triple transgenic mouse line ($n = 9$) was bred to express GCaMP6s in cortical excitatory neurons[66]. Mice ranging in age from 6 to 20 weeks of both sexes (four males and five females) were implanted with a head plate and cranial window and imaged starting >2 weeks after recovery from surgical procedures and up to 10 months after window implantation. The animals were housed on a 12 h light/dark cycle in cages of up to five animals before the implants, and individually after the implants. All animal procedures were approved by the Institutional Animal Care and Use Committee at the University of California, Santa Barbara.

**Surgical procedures**. All surgeries were conducted under isoflurane anesthesia (3.5% induction, 1.5−2.5% maintenance). Prior to incision, the scalp was infiltrated with lidocaine (5 mg/kg, subcutaneous) for analgesia and meloxicam (1−2 mg/kg, subcutaneous) was administered preoperatively to reduce inflammation. Once anesthetized, the scalp overlying the dorsal skull was sanitized and removed. The periosteum was removed with a scalpel and the skull was abraded with a drill burr to improve the adhesion of dental acrylic. A 4 mm craniotomy was made over the visual cortex (centered at 4.0 mm posterior, 2.5 mm lateral to Bregma), leaving the dura intact. A cranial window was implanted over the craniotomy and sealed first with silicon elastomer (Kwik-Sil, World Precision Instruments) then with dental acrylic (C&B-Metabond, Parkell) mixed with black ink to reduce light transmission. The cranial windows were made of two rounded pieces of coverglass (Warner Instruments) bonded with a UV-cured optical adhesive (Norland, NOA61). The bottom coverglass (4 mm) fit tightly inside the craniotomy while the top coverglass (5 mm) was bonded to the skull using dental acrylic. A custom-designed stainless

steel head plate (eMachineShop.com) was then affixed using dental acrylic. After surgery, mice were administered carprofen (5–10 mg/kg, oral) every 24 h for 3 days to reduce inflammation. The full specifications and designs for head fixation hardware can be found on the Goard lab website (https://goard.mcdb.ucsb.edu/resources).

Note that we performed glass prism implant surgeries on two of the mice[25] to record from L2-5 neurons in V1. In this work, we only performed analysis on L2/3 neurons.

**Two-photon imaging**. After >2 weeks' recovery from surgery, GCaMP6s fluorescence was imaged using a Prairie Investigator two-photon microscopy system with a resonant galvo scanning module (Bruker). Prior to two-photon imaging, epifluorescence imaging was used to identify the visual area being imaged by aligning to areal maps measured with widefield imaging. For fluorescence excitation, we used a Ti:Sapphire laser (Mai-Tai eHP, Newport) with dispersion compensation (Deep Sea, Newport) tuned to λ = 920 nm. For collection, we used GaAsP photomultiplier tubes (Hamamatsu). We used a 16×/0.8 NA microscope objective (Nikon) at 1× or 2× magnification, obtaining a square field of view with a width ranging from 414 to 828 μm. Laser power ranged from 40 to 75 mW at the sample depending on GCaMP6s expression levels. Photobleaching was minimal (<1%/min) for all laser powers used. A custom stainless-steel light blocker (https://goard.mcdb.ucsb.edu/resources) was mounted to the head plate and interlocked with a tube around the objective to prevent light from the visual stimulus monitor from reaching the PMTs. During imaging experiments, the polypropylene tube supporting the mouse was suspended from the behavior platform with high tension springs to reduce movement artifacts.

For imaging across multiple weeks, imaging fields on a given recording session were manually aligned based on visual inspection of the average map from the reference session recording, guided by stable structural landmarks such as blood vessels and neurons with high baseline fluorescence. Physical controls were used to ensure precise placement of the head plate and the visual stimulus screen relative to the animal, and data acquisition settings were kept consistent across sessions. Recordings were taken once every 7 ± 1 days for 5–7 weeks. To acclimate to head fixation and visual stimulus presentation, mice were head-fixed and presented the full series of visual stimuli for 1 to 2 full sessions prior to the start of their experimental run.

**Two-photon post-processing**. Images were acquired using PrairieView acquisition software and converted into TIF files. All subsequent analyses were performed in MATLAB (Mathworks) using custom code (https://goard.mcdb.ucsb.edu/resources). First, images were corrected for X-Y movement within each session by registration to a reference image (the pixel-wise mean of all frames) using two-dimensional cross-correlation. Next, to align recordings to the reference session, we used a semi-automated method similar to prior work[67,68]. First, anchor points were automatically generated from matching image features between average projections detected by the 'Speeded-Up Robust Features' (SURF) algorithm (Computer Vision Toolbox, Mathworks), and were manually corrected and added through visual inspection when necessary. These anchor points defined a predicted displacement vector field that would be used to map coordinates from one session to the other. For each coordinate, the predicted vector was defined by the average (weighted inversely by distance) of the vectors for all defined anchor points. This vector field was then applied to every frame of the recording to warp the coordinates of each image to the reference coordinate plane.

To identify responsive neural somata, a pixel-wise activity map was calculated using a modified kurtosis measure. Neuron cell bodies were identified using local adaptive threshold and iterative segmentation. Automatically defined ROIs were then manually checked for proper segmentation in a graphical user interface (allowing comparison to raw fluorescence and activity map images). To ensure that the response of individual neurons was not due to local neuropil contamination of somatic signals, a corrected fluorescence measure was estimated according to:

$$F_{corrected}(n) = F_{soma}(n) - \alpha(F_{neuropil}(n) - \bar{F}_{neuropil}) \qquad (1)$$

where $F_{neuropil}$ was defined as the fluorescence in the region <30 μm from the ROI border (excluding other ROIs) for frame $n$ (see Supplementary Fig. 8 for example neuropil signal traces). $\bar{F}_{neuropil}$ is $F_{neuropil}$ averaged over frames. $\alpha$ was chosen from [0, 1] to minimize the Pearson's correlation coefficient between $F_{corrected}$ and $F_{neuropil}$. Empirically, $\alpha$ is typically close to 1 and does not change significantly across weeks. The $\Delta F/F$ for each neuron was then calculated as:

$$\Delta F/F = (F_n - F_0)/F_0 \qquad (2)$$

Where $F_n$ is the corrected fluorescence ($F_{corrected}$) for frame $n$ and $F_0$ is defined as the mode of the corrected fluorescence density distribution across the entire time series.

To minimize potential artifacts introduced by misalignments of the imaging field across sessions, we manually inspected the average projection and pixel-wise activity maps underlying every defined ROI across all sessions. We assigned each ROI a quality rating based on its appearance and included only ROIs of sufficient quality in our analyses. Briefly, we defined ROI quality as follows: ROIs rated a quality of 4 or 5 were cells that were clearly present across sessions, and the cell

structure could be clearly resolved in both the average projection and activity map. ROIs rated a quality of 3 were also cells unambiguously tracked across sessions but had average maps that were often noisier than cells rated 4 or 5 (for example, they may be identifiable solely by their appearance on the activity map). ROIs rated a quality of 2 were either cells that were not well-tracked or were not unequivocally neuronal somata. ROIs rated a quality of 1 were cells that were not present on the reference session. Each ROI was also marked as either present or not present on each session. For our analysis, we only included ROIs which were presented on all the sessions and with a quality larger than 3.

**Visual stimuli.** All visual stimuli were generated with a Windows PC using MATLAB and the Psychophysics toolbox[69]. Stimuli used for two-photon imaging were presented on an LCD monitor ($17.5 \times 13$ cm, $800 \times 600$ pixels, 60 Hz refresh rate) positioned 5 cm from the eye at a horizontal tilt of 30° to the right of the midline and vertical tilt of 18° downward, spanning 120° (azimuth) by 100° (elevation) of visual space in the right eye.

For natural movie visual stimulation, we displayed a grayscale 30 s clip from *Touch of Evil* (Orson Wells, Universal Pictures, 1958) containing a continuous visual scene with no cuts (https://observatory.brain-map.org/visualcoding/stimulus/natural_movies). The clip was contrast-normalized and presented at 30 frames per second. We presented 30 repeats of the natural movie stimulus; each repeat started with 5 s of gray screen, followed by the 30 s of movie.

**Spiking episodes.** We first calculated deconvolved traces from $\Delta F/F$ using Suite-2p toolbox[33][2017]. For every neuron, we binarized the deconvolved trace by thresholding at 3 standard deviation above 0 to get inferred spikes. To calculate peri-stimulus time histogram (PSTH) for a given neuron, we first summed the inferred spikes across trials and smoothed them using Bayesian adaptive regression splines[70]. The spiking episode in each neuron was defined in the following steps. First, we found peaks with a prominence larger than 3 in the smoothed PSTH. Second, the full width at half maximum (FWHM) of the peaks defined the duration of spiking episodes in most cases. When the FWHM of neighboring peaks overlapped, the duration was defined by the difference between the start of the first peak and the end of the last peak.

**Nonnegative tensor decomposition with missing data.** We organized our data into a three-way tensor $\chi$ ($N \times T \times K$) and let $x_{ntk}$ represent the activity of neuron n at time t and trial k. Nonnegative TCA decomposes $\chi$ into a sum of $R$ rank-one tensors, where each rank-one tensor can be written as an outer product of three nonnegative vectors:

$$x_{ntk} \approx \sum_{r=1}^{R} w_n^r b_t^r a_k^r = \hat{x}_{ntk} \qquad (3)$$

Nonnegative TCA with missing values was fit to minimize the squared reconstruction error:

$||M * (\chi - \hat{\chi})||_F^2$ while $W \ge 0, B \ge 0, A \ge 0$

Here, $\hat{\chi}$ denotes the reconstructed data. $|| \cdot ||_F^2$ denotes the squared Frobenius norm of a tensor:

$$||\chi||_F^2 = \sum_{n=1}^{N} \sum_{t=1}^{T} \sum_{k=1}^{K} x_{ntk}^2 \qquad (4)$$

$M$ denotes a masking tensor with the same shape as $\chi$, and $\star$ denotes entrywise multiplication of two tensors. For fitting nonnegative TCA on $\Delta F/F$ data, we set $m_{ntk} = 0$ if $x_{ntk} < 0$, otherwise we set $m_{ntk} = 1$. Normalized reconstruction error is the squared reconstruction error normalized by $||M \star \chi||_F^2$.

Different from matrix decompositions, tensor decompositions are often unique[71]. However, when $R$ is large or $W$, $B$, $A$ have low rank, it could be difficult to optimize. To monitor this possibility, we calculated similarity between different TCA fitting results on the same dataset as described in[35]. We found that the similarity between fitting results is close to 1 for all the nonnegative TCA models reported in this work.

**Preprocessing of $\Delta F/F$ data.** $\Delta F/F$ data were normalized such that the averaged squared sum of $\Delta F/F$ traces over time equals to 1 for every neuron:

$$\sqrt{(\sum_{tk} x_{ntk}^2)/TK} = 1 \qquad (5)$$

This normalization step is crucial for ensuring TCA fitting is not biased by high firing rate neurons, since TCA is optimized to minimize the squared reconstruction error.

**Choice of the number of components in TCA.** We picked the number of TCA components such that they captured a significant amount of neural responses without over-fitting, checked with cross-validation as previously reported[35]. To perform cross-validation, we randomly masked out 50% of tensor entries in $\chi$. The remaining data was a training set and the masked-out data was a test set. We trained nonnegative TCA with missing values to fit the training set. And then we used the trained TCA model to fit the test set. As we increase the number of components in TCA, if the normalized reconstruction error of the test set went up, the TCA model would overfit the training set. As previously reported[35], TCA is unlikely to overfit, even with up to 70 components. For this paper, we chose 40

components for TCA, given that 40 component TCA captured a significant amount of neural responses without over-fitting (Supplementary Fig. 2).

**Isomap.** The instantaneous (temporal frequency: 10 Hz) population response $\Delta F/F$ of N recorded neurons is a point in an $N$-dimensional state space. Each axis in this state-space represents the activity of one neuron. A given trial of 35 s duration generates a discrete sequence (temporal frequency: 10 Hz) of 350 such points. The population activity from all trials (30 trials per recording session and six sessions) forms a cloud of 63,000 points in this $N$-dimensional state space. For the unsupervised transformation of the high-dimensional point cloud to a low-dimensional space, we ignored the association of a point to a given trial and to the time within the trial. We computed the Euclidean distance between all points, irrespective of the trial number and within-trial time. Based on the Euclidean distance we assigned 20 nearest neighbors to each point (choosing a higher number of nearest neighbors also works).

This step of nearest neighbor assignment is sensitive with respect to the existence of independent fluctuations of $\Delta F/F$ responses (i.e, independent noise). To discover meaningful structure in the population activity, we removed such independent noise. Rather than working with $\Delta F/F$ directly, we conducted the nearest neighbor assignment based on the "TCA-reconstructed $\Delta F/F$", from which the independent noise was removed.

By linking (edge) each point to its thus defined nearest neighbors, we translated the point cloud of population responses into a graph, i.e., a network of vertices (points) with edges (between a point and its nearest neighbor). The geodesic distance between two vertices in the graph is the distance of the shortest path connecting them. For our data set, the graph was described by the geodesic distance matrix of dimension $63,000 \times 63,000$.

Based on the pairwise geodesic distance between data points, we thus performed a transformation from the population responses in the $N$-dimensional state space to a space of lower dimensions. This isometric mapping method ("Isomap") was chosen to incorporate the presumed (but a priori unknown) manifold structure in the resulting transformation to a low-dimensional space. Isometric mapping preserves essential structure within the neuronal population responses. Note that the top $k$ eigenvectors of the geodesic distance matrix represent the coordinates (Isomap dimensions) in the new $k$-dimensional Euclidean space.

With all 63,000 data points successfully mapped into a state-space of $n$ dimensions, we recalled the assignment of each point to a given trial and to the time within the trial. This temporal sequence of data points formed the trajectory of population activity for a given trial in this low-dimensional space.

Shuffled data with circularly shifted responses across trials have much higher intrinsic dimensions than original data. Due to the curse of dimensionality[72] and the smoothness of shuffled responses, we need to define a larger neighborhood size for Isomap to reveal a robust topology of the neural manifold in this case. Thus, we chose 100 nearest neighbors for Isomap for shuffled data.

**Spline parameterization for unsupervised decoding (SPUD).** We used the SPUD algorithm described in[39]. We fitted the manifolds with piecewise linear curves. We chose to fit a curve $L(y)$ with ten knots to the data points $x_i$ embedded in the two-dimensional spaces by Isomap. Initially, the positions of knots were determined by $K$-means clustering centroids of the data points. Each knot was connected to the other knot with the highest data point density in between to form the initial curve. Then, positions of the knots were iteratively optimized to minimize ($\Sigma i ||(L(y) - x_i)|| |L(y)|$), where $||(L(y) - x_i)||$ is the Euclidean distance between the ith data point and the nearest point on the curve, and $|L(y)|$ is the length of the curve.

We picked a random origin on the curve and assigned coordinates from 0 to 1 to the point on the curve. The coordinate of each data point $x_i$ was decoded as the coordinate of its nearest point on the curve. We shifted or flipped the coordinates of the data points to minimize the mean squared error between the decoded coordinates and the rescaled actual time in the movie (rescaled to (0,1]). The decoded time for a given data point was set to the resulting coordinate scaled up to (0, 35) seconds.

For cross-validation, we randomly picked 80% of the instantaneous population activity as the training set (distributed across all the weeks), and the remaining 20% as the test set. We fit the spline to data from the training set and evaluated the decoding performance using data from the test set.

Note that the neural manifold for shuffled data often did not have a perfect ring structure (Supplementary Fig. 7a). The SPUD would fail without carefully choosing the positions of initial knots. For a fair quantitative comparison between original and shuffled data (Supplementary Fig. 7b), we chose ten trial-averaged projected instantaneous population activity evenly distributed in time as the initial knots for the shuffled data analysis.

**The variance of population activity along/perpendicular to the coding direction.** We calculated the variance of population activity along or perpendicular to the coding direction based on the coordinates of the projected instantaneous population activity along or perpendicular to the spline direction identified by SPUD. The variance of population activity reported in Figs. 6, 7 was calculated based on projected population activity in the first two Isomap dimensions.

**Radius of points on the manifold**. We quantified the shape of the neural manifold for original and shuffled data by calculating the distance from each dot representing instantaneous population activity to the center of the manifold. The Center of the manifold was calculated as averaged coordinates across all the points. Empirically, the center was close to the origin.

**Reporting summary**. Further information on research design is available in the Nature Research Reporting Summary linked to this article.

## Data availability
Most of the hardware designs can be found on Michael Goard's lab website (https://goard.mcdb.ucsb.edu/resources). Raw data analyzed in this study have been deposited in the Dryad https://doi.org/10.25349/D9M606.

## Code availability
We used tools for fitting TCA in https://github.com/ahwillia/tensortools. We used code available from https://fietelab.mit.edu/code/ for SPUD. A sample dataset and a Jupyter notebook for reproducing some of the main figures are available from Supplementary Software. All the other code used for analysis is available upon request to the corresponding author.

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

## Acknowledgements

This work was supported by the following grants: Whitehall Foundation #20121221 (to R.W.), NSF CRCNS #1308159 (to R.W.), NIH R00 MH104259 (to M.J.G.), Whitehall Foundation #20181228 (to M.J.G.), NSF NeuroNex #1707287 (to M.J.G.), and R01 NS121919 (to M.J.G.).

## Author contributions

J.X., T.D.M., M.J.G. and R.W. conceived and designed research; T.D.M. performed experiments; J.X. analyzed data; J.X., T.D.M., M.J.G. and R.W. interpreted results of experiments; J.X. prepared figures; J.X. drafted the paper; J.X., T.D.M., M.J.G. and R.W. edited and revised paper; J.X., T.D.M., M.J.G. and R.W. approved the final version of the paper.

## Competing interests

The authors declare no competing interests.
