## [Peer Review File · Nature Communications]

Stable representation of a naturalistic movie emerges from episodic activity with gain variabilityREVIEWER COMMENTS

Reviewer #1 (Remarks to the Author):

The authors take an unsupervised approach to exploring the structure of population responses and their trial-to-trial and session-to-session variability. The primary claim of the manuscript is that although visual cortex responses vary from trial to trial and from session to session, a stable low-dimensional representation persists over time.

The analysis of trial-to-trial variability in neuronal responses using TCA in Figure 3 is quite interesting. It reveals that population responses include relatively stable components (the top cluster in Figure 3c) and other components that appear to come and go over the course of weeks.

The following analyses examining stability of population representations are less compelling. My specific concerns are described below. More generally, it not clear to me whether the authors conclusions simply follow from the fact that the analyzed neuronal population responds to the natural movie stimulus and at least some of the neurons display some consistency in their responses over sessions. I could not figure out what kind of data would be incompatible with the authors' observation of "a stable one dimensional representation of the time in the natural movie". It seems that violating this would require visual cortex neurons to respond drastically differently from session to session, in a way that would be fundamentally at odds with the with much simpler analyses of stability of V1 responses. It is clear from the presented examples that neurons' firing patterns do change over time, but the time points of the movie where they are most active often remain the same. Decoding the time during the trial should be straightforward as long as enough such neurons are included in the population.

1. Perhaps I am missing some subtlety here but it seems that the ring structure of the manifold in Figure 4 is a simple consequence of the periodic structure of stimulus presentation (repeats of the same stimulus). This basic point – the fact the population responses are dominated by visual responses to a periodic visual stimulus – is also at the core of concerns about interpretation of the follow-up analyses.

2. No details are provided on how the data were divided into the training and test set in Figure 5, leaving me to assume that the training data is distributed across all weeks. It is therefore entirely expected that timing can be decoded from the test data. It is not at all clear to me what this analysis says specifically about week-to-week stability of neuronal responses. Yes, it is clear that there is some stable representation that persists across weeks, as one would expect, but that much is obvious from simply looking at single cell responses.

3. I could not figure out how exactly is the variance of population activity in Figure 6c and later figures calculated and am assuming that it is computed based on coordinates in the two-dimensional Isomap projection. More details in the Methods would be helpful. Since the spline dimension corresponds to time in the movie stimulus, it would be very surprising if the bulk of the variability lay along this dimension. That would imply that on different trials or recording sessions the visual cortex neuronal population responds to events in the movie stimulus either substantially earlier or later than they occur.

The authors seem to concede that point later, in the analysis of shuffled data in Figure 7, stating that "the fact that episodic activity is precise in time across trials (Fig. 2c) alone is sufficient for constraining neural variability to the direction perpendicular to the stimulus encoding direction" (lines 337-339). Isn't the precision of episodic activity in time across trials a simple consequence of the fact that the authors are analyzing the activity of a visually responsive population? The fact that the radius of the manifold decreases after the shuffling procedure that preserves trial structure (Figure 7g) suggests that this doesn't fully explain the authors' findings but this point is not explored further.

In addition, less stable components of population activity may be captured in higher dimensions of the

Isomap projection. As the Isomap picks up on the trial structure in the data, and captures it in its top two dimensions (Figure 4c), components of activity that vary across trials and sessions may be discarded in a two-dimensional projection.

Minor issues:

1. I urge the authors to use caution when referring to “spike timing” estimated by inferring spikes from GCaMP6s fluorescence. This phrasing invokes debates on neural coding, where spike timing is considered on millisecond time scales, far beyond the temporal resolution of the calcium imaging. The results of deconvolution of fluorescence traces should not be literally thought of as spikes and it would be more conservative to refer to “event rates” rather than “spike rates” in Figure 2.

2. Line 330 – typo, “spine” should be “spline”.

3. Line 636 – is “force” meant to be “fair”?

Reviewer #2 (Remarks to the Author):

Xia and colleagues performed an impressive analysis of visual responses across weeks and found that, although responses changed across weeks, the underlying latent dimensions of the responses are preserved. The computational analyses and findings are novel. However, there is not enough technical detail to ensure that the neurons are indeed tracked across weeks. I suggest the authors perform a few technical controls to strengthen the findings of the paper.

Major:

* Given the study is focused on multiday imaging, more validation of the imaging technique needs to be shown. For instance, can a quantification of the correlation of the mask on the max projection image across days be computed and some threshold based on that used? Or is there a different measure for cell inclusion across days that I missed? Also, is the alpha value for neuropil fixed across days? If you are in the same place in Z across days the estimated alpha value should not change too much.

* Does binarizing the spike deconvolved data influence your results? Given the non-linearity of GCaMP6s, binarizing the traces will result in selection of primarily bursting periods. Binarizing the traces will also likely increase the trial-to-trial variability of the signal.

* Interesting manifold embedding analyses. What is your interpretation of the ISOMAP being circular? Is there some structure in the movie that would underlie that structure?

Minor:

* Line 78 “in a”

* Do you observe these “gain” changes in the drifting gratings responses as well? It would support your hypothesis further if so.

* What are the mechanisms underlying this drift? Does the neuropil signal (aka the input, sort of) show this type of drift across weeks?

Reviewer #3 (Remarks to the Author):

This is a very interesting paper showing that although neuronal responses to a short strip of a natural stimulus movie vary a lot from week to week in terms of firing rate, they vary much less in terms of the time-point at which they occur with respect to the timeline of the movie. This allowed the authors to employ a dimensionality reduction algorithm to extract a direction in population activity space along which population responses remain relatively invariant and perpendicular to which directions reflect non-coding modulations. The work is novel, of general interest, and contributes significantly to the field. Below I list some comments to address:

Major Comments

1) The authors report that coordinated gain changes are a key to maintaining the stability of stimulus representation in area V1. This is reminiscent to work by Lee et al (J Neurosci, 39(9):1671-1687, 2019) who in a different context noted that the gain factor across V1 neuronal populations was more uniform for internally driven activity modulations as opposed to external modulations driven by changes in stimulus contrast, which were more heterogeneous across neurons. I wonder whether the authors could comment about the relation of their analysis to this work.

Indeed, it would be interesting to show directly how the distribution of gain factors across the population of neurons for the same episode of firing changes over time. This could be compared to how the distribution of gains across neurons vary between different episodes of firing (occurring within the same trials).

2) Several behavioral parameters of the state of the animal are known to affect neuronal firing and gain controls, and it is important to consider whether they can explain the variability reported here. For example what is the effect of the state of locomotion of the animal, pupillary size, eye movement range etc on the trajectories and isomap dimensions identified. If data are not available, please discuss/model the possible impact of these factors.

3) The neuropil correction implemented removes the pearson correlation between the neuropil signal and the nearby neuron. This procedure can potentially eliminate real firing events from the time series of the neuron, if these lie "parallel" to the neuropil modulation. What effect, if any, does this have on the trajectories identified? Are the results stable to implementing a different neuropil correction strategy (for example using the factor 0.7 suggested by Bonin et al.)? how different are the results without the neuropil correction? Plotting some of the raw data traces from the neurons and nearby neuropil may help and it would also give a direct impression of the quality of the data.

4) Similar to #3, it is good to discuss in more detail the potential effect of the normalization condition implemented during data preprocessing (setting the L2 norm of df/f of each neuron to 1). Would an arguably more realistic normalization method, which would normalize for aggregate activity across a population of neurons impact the conclusions of the study?

5) Given the low number of isomap dimensions along which useful information is conveyed, how do you envision the full representation of the stimulus being encoded? For example, the current isomap projection appears to address more a question along the lines of "when" a stimulus event occurred, versus "what" the stimulus actually was at that time. How do you expect the number of isomap dimensions to scale with the number of neurons recorded from?

Minor Comments: 1) Ensure abbreviations are defined before being used in the manuscript. Improve the English grammar.

2) formula typo in line 541

We thank the reviewers for their constructive criticism. Changes to the manuscript made in response to reviewer comments were marked in red. For the reviewers' convenience, we respond (in blue text) to each comment individually.

Reviewer #1 (Remarks to the Author):

The authors take an unsupervised approach to exploring the structure of population responses and their trial-to-trial and session-to-session variability. The primary claim of the manuscript is that although visual cortex responses vary from trial to trial and from session to session, a stable low-dimensional representation persists over time.

The analysis of trial-to-trial variability in neuronal responses using TCA in Figure 3 is quite interesting. It reveals that population responses include relatively stable components (the top cluster in Figure 3c) and other components that appear to come and go over the course of weeks.

The following analyses examining stability of population representations are less compelling. My specific concerns are described below. More generally, it is not clear to me whether the authors' conclusions simply follow from the fact that the analyzed neuronal population responds to the natural movie stimulus and at least some of the neurons display some consistency in their responses over sessions. I could not figure out what kind of data would be incompatible with the authors' observation of "a stable one dimensional representation of the time in the natural movie". It seems that violating this would require visual cortex neurons to respond drastically differently from session to session, in a way that would be fundamentally at odds with the much simpler analyses of stability of V1 responses. It is clear from the presented examples that neurons' firing patterns do change over time, but the time points of the movie where they are most active often remain the same.

Decoding the time during the trial should be straightforward as long as enough such neurons are included in the population.

We thank the reviewer for the insightful comments and pointing out this apparent confusion about our analysis on the population representation. Stable representation of the time in the natural movie requires not only some neurons display some consistency in their responses over sessions, but also that 1) different groups of neurons be activated at different timepoints in the movie and 2) the activation of episodic activity tile across the entire duration of the movie (no silent period during the movie).

1) and 2) both require coordination between episodic activity from different neurons. If the same group of neurons were activated at different timepoints in the movie or if we had a silent period during the movie without activation of any neurons, then not all the time points in the movie would be uniquely represented in the neural space and we would see a

collapsed ring structure for the neural manifold as shown in Fig 7d (from components with shuffled factors).

Decoding time during the trial given reliable neurons using supervised methods is straightforward. However, here instead of asking whether time in the movie is stably represented, we asked what information is stably represented in the neural space by using an unsupervised method. The attempt to parameterize the neural manifold has the potential of revealing stable representation of other stimulus-related variables. Thus, we think it is useful to report the neural manifold analysis.

Motivated by the reviewer's concern, we have clarified this point in the text (Line 365-370) as follows: "The collapsed ring structure of shuffled data showed that stable representation of the time in the natural movie requires not only some neurons display reliable responses over sessions (shuffled data also have neurons with reliable responses), but also coordination between episodic activity."

1. Perhaps I am missing some subtlety here but it seems that the ring structure of the manifold in Figure 4 is a simple consequence of the periodic structure of stimulus presentation (repeats of the same stimulus). This basic point – the fact the population responses are dominated by visual responses to a periodic visual stimulus – is also at the core of concerns about interpretation of the follow-up analyses.

We agree with the reviewer that the periodic structure of the stimulus presentation is necessary for forming the ring structure of the manifold. However, the fact that neural trajectories from different trials across weeks are well-aligned are also necessary for forming the ring structure, which is not guaranteed given the neural variability we observed across weeks. Without the periodic structure of stimulus, we expect to see a line structure. To avoid confusion, we add a sentence at Line XXX to clarify: "Note that the ring structure of the manifold arises from the looped trial structure of the visual stimulus. If the stimulus were repeated but not looped in time, i.e., interleaved with different stimuli between trials, we would expect to see a line structure for the manifold with well-aligned trajectories."

The shape of the manifold also depends on the similarity between neural responses to movie scenes on different frames. For example, if visual stimuli consisted of drifting gratings with different orientations (spanning 360 degrees) shown in fixed order, then the neural manifold would form two aligned but non-overlapping rings (see Figure S1i in (Stringer, Michaelos and Pachitariu, 2019)) due to the similarity between neural responses to gratings with opposite drifting directions (orientations of gratings differ by 180 degrees).

2. No details are provided on how the data were divided into the training and test set in Figure 5, leaving me to assume that the training data is distributed across all weeks. It is therefore entirely expected that timing can be decoded from the test data. It is not at all

clear to me what this analysis says specifically about week-to-week stability of neuronal responses. Yes, it is clear that there is some stable representation that persists across weeks, as one would expect, but that much is obvious from simply looking at single cell responses.

Thank you for pointing out this missing information. We added a description of how data was divided into the training and test set in the methods section (See Line 675-680): “For cross-validation, we randomly picked 80% of the instantaneous population activity as the training set (distributed across all the weeks), and the remaining 20% as the test set. We fit the spline to data from the training set and evaluated the decoding performance using data from the test set.”

The reported analysis based on a random split of training and test data across all the weeks in Figure 5 illustrated that there existed a fixed direction (the spline direction) that allowed us to readout time in the movie across all the weeks. In response to the reviewer's comment, we trained SPUD decoders on data from odd trials in week 1 and tested its performance using data from even trials in week 1 and data from later weeks. We found that the decoding errors pooled from later weeks were not significantly different from the decoding errors for week 1 ($p = 0.11$, Mann-Whitney U test; Fig. a, b). This additional analysis showed that week-to-week variability does not affect the coding of time in the movie.

a. Decoding errors of SPUD trained on data from odd trials in week 1 and tested on data from even trials in week 1 and trials from other weeks. Different colors correspond to different imaging fields.

- b. Decoding errors averaged over imaging fields for SPUD trained on data from odd trials in week 1. The decoding errors pooled from later weeks were not significantly different from the decoding errors for week 1 across imaging fields ($p = 0.11$, Mann-Whitney U test).

Motivated by the reviewer's comment, we added this figure to supplementary figure 5. And we added sentences to the results section at Line 250-255: "To further investigate the stability of neural representation of time in the trial, we also trained SPUD on neural data from odd trials in week 1 and tested its performance on neural data from even trials in week 1 and trials from other weeks. The decoding errors pooled from later weeks were not significantly different from the decoding errors for week 1 across imaging fields ($p = 0.11$, Mann-Whitney U test; Supplementary Fig. 5c, d). This additional analysis showed that week-to-week variability does not affect the coding of time in the movie."

3. I could not figure out how exactly is the variance of population activity in Figure 6c and later figures calculated and am assuming that it is computed based on coordinates in the two-dimensional Isomap projection. More details in the Methods would be helpful.

Since the spline dimension corresponds to time in the movie stimulus, it would be very surprising if the bulk of the variability lay along this dimension. That would imply that on different trials or recording sessions the visual cortex neuronal population responds to events in the movie stimulus either substantially earlier or later than they occur.

The authors seem to concede that point later, in the analysis of shuffled data in Figure 7, stating that "the fact that episodic activity is precise in time across trials (Fig. 2c) alone is sufficient for constraining neural variability to the direction perpendicular to the stimulus encoding direction" (lines 337-339). Isn't the precision of episodic activity in time across trials a simple consequence of the fact that the authors are analyzing the activity of a visually responsive population? The fact that the radius of the manifold decreases after the shuffling procedure that preserves trial structure (Figure 7g) suggests that this doesn't fully explain the authors' findings but this point is not explored further.

In addition, less stable components of population activity may be captured in higher dimensions of the Isomap projection. As the Isomap picks up on the trial structure in the data, and captures it in its top two dimensions (Figure 4c), components of activity that vary across trials and sessions may be discarded in a two-dimensional projection.

We thank the reviewer for pointing out the omission of methods description. Yes, the variance of population activity is computed based on coordinates in the first two Isomap dimensions. We added the corresponding description to Line 690-695: "We calculated the

variance of population activity along or perpendicular to the coding direction based on the coordinates of the projected instantaneous population activity along or perpendicular to the spline direction identified by SPUD. The variance of population activity reported in Fig 6, 7 was calculated based on projected population activity in the first two Isomap dimensions.”

We respectfully disagree with the reviewer that visual responsiveness equals reliable spike timing. A priori, the timing of the episodic activity does not have to be precise in time across weeks. Based on previous reports in PPC (Driscoll et al., 2017) and hippocampus (Ziv et al., 2013), episodic activity with shifted timing across weeks is a plausible hypothesis as well. Without measuring and analyzing the chronic neural activity in V1, we wouldn't know whether the timing of the episodic activity is stable or not.

We agree with the reviewer that less stable components of population activity are captured in the higher dimensions of the Isomap projection, as shown by Fig. 4c. And those variability captured by the higher Isomap dimensions are likely to be perpendicular to the coding direction as well, as the spline lies in the subspace perpendicular to those higher dimensions.

Minor issues:

1. I urge the authors to use caution when referring to “spike timing” estimated by inferring spikes from GCaMP6s fluorescence. This phrasing invokes debates on neural coding, where spike timing is considered on millisecond time scales, far beyond the temporal resolution of the calcium imaging. The results of deconvolution of fluorescence traces should not be literally thought of as spikes and it would be more conservative to refer to “event rates” rather than “spike rates” in Figure 2.

We thank the reviewer for pointing this out. We avoided mentioning “spike timing” in the manuscript. We changed “inferred spikes” to “inferred spiking activity”, and “spike rates” to “inferred spike rates” in the figures and results section, in line with notations from previous works (Wei et al., no date; Theis et al., 2016; Zhang et al., 2018). We also add a caveat of interpreting the inferred spiking activity by referring to (Huang et al., 2021) at Line 100-105 to avoid confusion: “Note that the inferred spiking activity might correspond to bursts of spikes instead of a single spike due to limitations of calcium imaging (Huang et al., 2021).”

We didn't choose to refer to “inferred spikes” as “events”, because “events” sometimes refer to calcium events calculated from $\Delta F/F$ traces without performing deconvolution (Peters, Chen and Komiyama, 2014; Ma et al., 2020; Marks and Goard, 2020).

2. Line 330 – typo, “spine” should be “spline”.

We thank the reviewer for catching this mistake. The typo has been fixed.

3. Line 636 – is “force” meant to be “fair”?

We thank the reviewer for catching this mistake. The typo has been fixed.

Reviewer #2 (Remarks to the Author):

Xia and colleagues performed an impressive analysis of visual responses across weeks and found that, although responses changed across weeks, the underlying latent dimensions of the responses are preserved. The computational analyses and findings are novel. However, there is not enough technical detail to ensure that the neurons are indeed tracked across weeks. I suggest the authors perform a few technical controls to strengthen the findings of the paper.

Major:

* Given the study is focused on multiday imaging, more validation of the imaging technique needs to be shown. For instance, can a quantification of the correlation of the mask on the max projection image across days be computed and some threshold based on that used? Or is there a different measure for cell inclusion across days that I missed? Also, is the alpha value for neuropil fixed across days? If you are in the same place in Z across days the estimated alpha value should not change too much.

We thank the reviewer for pointing out the missing information. More details of validation of the imaging technique are reported in a related manuscript (Marks and Goard, 2020). Please see the text below for a detailed description of cell inclusion criteria (Line 635-645 from (Marks and Goard, 2020)):

“To minimize potential artifacts introduced by misalignments of the imaging fields across sessions, we manually inspected the average projection and pixel-wise activity maps underlying every defined ROI across all sessions. We assigned each ROI a quality rating based on its appearance and included only ROIs of sufficient quality in our analyses (threshold quality of 3 unless indicated otherwise). Briefly, we defined ROI quality as follows: ROIs rated a quality of 4 or 5 were cells that were clearly present across sessions, and the cell structure could be clearly resolved in both the average projection and activity map. ROIs rated a quality of 3 were also cells unambiguously tracked across sessions but had average maps that were often noisier than cells rated 4 or 5 (for example, they may be identifiable solely by their appearance on the activity map). ROIs rated a quality of 2 were either cells that were not well-tracked or were not unequivocally neuronal somata. ROIs rated a quality of 1 were cells that were not present on the reference session. Each ROI was also marked as either present or not present on each session.”

In response to the reviewer’s comment, we adapted the paragraph above and added it to our methods section at Line 535-550.

The alpha value for neuropil is adapted across days to minimize the Pearson's correlation between $F_{corrected}$ and $F_{neuropil}$ (alpha was chosen from [0, 1]), however, alpha value does not change much for the same neuron across weeks (Fig a), and distribution of alpha values also does not change significantly across weeks (Fig b; $p > 0.05$ for any pairs of weeks, Mann-Whitney U test).

- a. α for all the neurons across weeks in one imaging field.
- b. Distribution of α in one imaging field. Different colors indicate distributions from different weeks.

In response to the reviewer's comment, we added sentences in the methods at Line 525-530: "Empirically, α is typically close to 1 and does not change significantly across weeks."

* Does binarizing the spike deconvolved data influence your results? Given the non-linearity of GCaMP6s, binarizing the traces will result in selection of primarily bursting periods. Binarizing the traces will also likely increase the trial-to-trial variability of the signal.

For clarification, we only used the binarized deconvolved data for results shown in Figure 2, and we performed analysis on $\Delta F/F$ for results shown in other figures. Thus, our results in other figures will not be affected. We chose to binarize the deconvolved data because we would like to quantify the stability of the timing of spiking episodes. We agree with the reviewer that binarizing the trace will likely increase the variability in spike rates across weeks, however, it would not change our conclusion in Figure 2 qualitatively.

*Interesting manifold embedding analyses. What is your interpretation of the ISOMAP being circular? Is there some structure in the movie that would underlie that structure?

The circular manifold is due to the looped trial structure of the visual stimulus. Without the looped structure of the stimulus, we expect to see a line structure. For clarification, we added sentences in the results section (See Line 215): “Note that the ring structure of the manifold arises from the looped trial structure of the visual stimulus. If the stimulus were repeated but not looped in time, i.e., interleaved with different stimuli between trials, we would expect to see a line structure for the manifold with well-aligned trajectories.”

The shape of the manifold also depends on the similarity between neural responses to movie scenes on different frames. For example, if visual stimuli consisted of drifting gratings with different orientations (spanning 360 degrees) shown in fixed order, then the neural manifold would form two aligned but non-overlapping rings (see Figure S1i in (Stringer, Michaelos and Pachitariu, 2019)) due to the similarity between neural responses to gratings with opposite drifting directions (orientations of gratings differ by 180 degrees).

Minor:

* Line 78 “in a”

We thank the reviewer for catching this mistake. The typo has been fixed.

* Do you observe these “gain” changes in the drifting gratings responses as well? It would support your hypothesis further if so.

Yes, we did observe gain changes in the drifting grating responses for some neurons (see example neuron shown in figure 1F in (Marks and Goard, 2020), its response to drifting grating with orientation 150° change in gains across weeks). However, for drifting grating responses, the proportions of episodic activity that change in gains are smaller than that of the naturalistic movie responses. (see figure 2D in (Marks and Goard, 2020))

* What are the mechanisms underlying this drift? Does the neuropil signal (aka the input, sort of) show this type of drift across weeks?

The mechanism underlying gain variations we observed in the episodic activity is an interesting topic to study in future. At the neural circuit level, there are several possible mechanisms that could lead to the observed drift. First, the turnover of boutons and dendritic spines in V1 at the baseline condition (Hofer et al., 2009; Holtmaat and Svoboda, 2009) would cause changes in the recurrent inputs to single neurons. Second, potential changes in the feedforward connectivity from LGN to V1 or drift in LGN responses would cause drift in the feedforward inputs to single neurons. Third, slowly varying top-down inputs related to visual information processing could contribute to the drift as well. Model investigations and simultaneous chronic recordings from LGN or high-order visual areas would be helpful for distinguishing contributions from these potential mechanisms in the future.

Motivated by the reviewer's question, we adapted the paragraph above and added it to the discussion at Line 430-440.

It is an interesting idea to relate neuropil signals to the input to neurons. However, it is hard to interpret the neuropil signal due to the multitude of signal sources. For example, the neuropil signals may consist of fluorescence signals from different dendrites or axons across weeks. To address the reviewer's question, we plotted the neuropil signal across weeks for two example ROIs (see figure below). In general, the neuropil lacked obvious episodic activity, and displayed qualitatively different and less drift compared to the ROIs.

a. $\Delta F/F$ of one example neuron, neuron 1.

- b. $\Delta F/F_{neuropil}$ of neuron 1.
- c. $\Delta F/F$ of another example neuron, neuron 2.
- d. $\Delta F/F_{neuropil}$ of neuron 2.

Reviewer #3 (Remarks to the Author):

This is a very interesting paper showing that although neuronal responses to a short strip of a natural stimulus movie vary a lot from week to week in terms of firing rate, they vary much less in terms of the time-point at which they occur with respect to the timeline of the movie. This allowed the authors to employ a dimensionality reduction algorithm to extract a direction in population activity space along which population responses remain relatively invariant and perpendicular to which directions reflect non-coding modulations. The work is novel, of general interest, and contributes significantly to the field. Below I list some comments to address:

Major Comments

1) The authors report that coordinated gain changes are a key to maintaining the stability of stimulus representation in area V1. This is reminiscent to work by Lee et al (J Neurosci, 39(9):1671-1687, 2019) who in a different context noted that the gain factor across V1 neuronal populations was more uniform for internally driven activity modulations as opposed to external modulations driven by changes in stimulus contrast, which were more heterogeneous across neurons. I wonder whether the authors could comment about the relation of their analysis to this work.

Indeed, it would be interesting to show directly how the distribution of gain factors across the population of neurons for the same episode of firing changes over time. This could be compared to how the distribution of gains across neurons vary between different episodes of firing (occurring within the same trials).

We thank the reviewer for this insightful comment. It would be interesting if we could distinguish between different modulation mechanisms based on the distribution of gain changes. One important distinction between our results and the work by Lee et al is that we defined gain changes in the units of spiking episodes instead of single neurons. The gain change distribution across spiking episodes is indeed heterogeneous (see figure below) and showed a skewed distribution similar to that of the gain modulations driven by stimulus contrast levels (Fig 4A in Lee et al).

Here Gain change is defined as the ratio between inferred spike rates (ISR) during the same spiking episodes of one week and of its previous week. We showed distribution of gain changes across spiking episodes from an example imaging field (same field as shown in Fig 2 in the manuscript). Different colors indicate gain changes between different adjacent week pairs. We only include spiking episodes whose ISR is non-zero during week i to avoid infinite gain changes.

The trial factors from our TCA analysis identified gains for the same episode shared across neurons, however, in our case, the gain changes for the same episode are the same across neurons (gain changes are summarized by the same trial factor across neurons), thus we didn't plot the distribution of the gain changes across neurons for the same episode.

Motivated by the reviewer's comment, we added sentences to the discussion at Line 430-435: "Previous studies (Niell and Stryker, 2010; Vinck et al., 2015; Lee, Park and Smirnakis, 2019) suggest behavioral variables such as locomotion and arousal could lead to gain modulations in single neural activity. However, the reported modulations by behavioral variables were homogeneous across neurons (Lee, Park and Smirnakis, 2019) and whether changes in behavioral variables would contribute the heterogeneous gain modulations across episodic activity within the same neuron (Fig. 2e, f) is not clear."

2) Several behavioral parameters of the state of the animal are known to affect neuronal firing and gain controls, and it is important to consider whether they can explain the variability reported here. For example what is the effect of the state of locomotion of the animal, pupillary size, eye movement range etc on the trajectories and isomap dimensions identified. If data are not available, please discuss/model the possible impact of these factors.

We agree with the reviewer that it would be interesting to discuss the impact of behavior variables on the neural trajectories identified by Isomap. In general, we found that the radius of the projected instantaneous population activity is correlated with the population mean responses (see figure below and Supplementary Fig. 4b). As shown by the previous work

(Niell and Stryker, 2010; Vinck et al., 2015), population mean responses could be affected by behavioral variables such as locomotion and arousal. Thus, changes in those behavioral variables may modulate the neural trajectories in the radial direction by affecting the population mean responses.

Motivated by the reviewer's comment, we added sentences to the discussion at Line 430-435: "Previous studies (Niell and Stryker, 2010; Vinck et al., 2015; Lee, Park and Smirnakis, 2019) suggest behavioral variables such as locomotion and arousal could lead to gain modulations in single neural activity. However, the reported modulations by behavioral variables were homogeneous across neurons (Lee, Park and Smirnakis, 2019) and whether changes in behavioral variables would contribute the heterogeneous gain modulations across episodic activity within the same neuron (Fig. 2e, f) is not clear."

3) The neuropil correction implemented removes the Pearson correlation between the neuropil signal and the nearby neuron. This procedure can potentially eliminate real firing events from the time series of the neuron, if these lie "parallel" to the neuropil modulation. What effect, if any, does this have on the trajectories identified? Are the results stable to implementing a different neuropil correction strategy (for example using the factor 0.7 suggested by Bonin et al.)? How different are the results without the neuropil correction? Plotting some of the raw data traces from the neurons and nearby neuropil may help and it would also give a direct impression of the quality of the data.

We thank the reviewer for pointing out this subtlety of data processing. In response to the reviewer's suggestion, we recalculated the $\Delta F/F$ responses from the example imaging field using the correction factor 0.7. As shown below, $\Delta F/F$ responses calculated using both neuropil correction strategies are similar to each other (Fig a, b). Isomap trajectories are qualitatively the same under the proposed neuropil correction (Fig c).

- a. Fluorescence traces for one example neuron. First row: $\Delta F/F$ calculated using the adapted α across weeks, for this neuron, $\alpha = 0.88$ for trials from week 5 (trial 120-150) and $\alpha = 1$ for the other trials. Second row: $\Delta F/F$ calculated using $\alpha = 0.7$ for all the trials. Third row: $\Delta F/F$ calculated without neuropil correction. Fourth row: $\Delta F/F$ of the neuropil signal.
- b. $\Delta F/F$ calculated with different neuropil correction methods for the example neuron.
- c. Isomap trajectories calculated from $\Delta F/F$ with different neuropil correction methods (left: adapted α ; middle: $\alpha = 0.7$; right: no neuropil correction).

4) Similar to #3, it is good to discuss in more detail the potential effect of the normalization condition implemented during data preprocessing (setting the L2 norm of df/f of each neuron to 1). Would an arguably more realistic normalization method, which would normalize for aggregate activity across a population of neurons impact the conclusions of the study?

We thank the reviewer for pointing out this additional subtlety of data processing. Just for clarification, the normalization is only applied for results in Figure 3-7, and it will not affect the results we showed in Figure 1-2. In general, choosing different normalization methods will not impact the conclusions of the study qualitatively. Normalization will change the scale of single neuronal responses to some extent, but it will not change the drifting pattern of single neuronal responses. For the neural manifold identified by Isomap, choosing different normalization methods will slightly affect how trajectories are twisted, but we would still see well-aligned neural trajectories across weeks (see figure below).

5) Given the low number of isomap dimensions along which useful information is conveyed, how do you envision the full representation of the stimulus being encoded? For example, the current isomap projection appears to address more a question along the lines of "when" a stimulus event occurred, versus "what" the stimulus actually was at that time. How do you expect the number of isomap dimensions to scale with the number of neurons recorded from?

Searching for the full representation of the stimulus is indeed an interesting topic, yet it is beyond the scope of the current manuscript. The stimulus used here (a naturalistic movie clip) is a high dimensional variable, it is not clear how we can parameterize the neural manifold in relation to the stimulus. In the future, it may be helpful to design low-dimensional parameterized visual stimuli and check how those visual-related parameters are encoded on the neural manifold. Moreover, to study encoding of “what” information, it would be necessary to disentangle “when” and “what” variables in the experimental design. For example, it could be helpful to show parameterized visual stimuli randomly varying in time without the looped trial structure.

If one intended to decode time in the trial from the neural activity, the number of Isomap dimensions needed is only 2 as shown in Fig 5, and it does not scale the number of neurons as long as we have enough number of neurons (empirically more than 100 neurons would be enough). In general, the number of neurons needed for decoding will also scale up with the dimensionality of the encoded variables (Gao and Ganguli, 2015).

Minor Comments: 1) Ensure abbreviations are defined before being used in the manuscript. Improve the English grammar.

Thanks for pointing out this mistake. We've checked throughout the manuscript and added definitions before abbreviations.

2) formula typo in line 541

We thank the reviewer for catching this mistake. The typo has been fixed.

References

Driscoll, L. N. *et al.* (2017) 'Dynamic Reorganization of Neuronal Activity Patterns in Parietal Cortex', *Cell*, 170(5), pp. 986–999.e16.

Gao, P. and Ganguli, S. (2015) 'On simplicity and complexity in the brave new world of large-scale neuroscience', *Current opinion in neurobiology*, 32, pp. 148–155.

Hofer, S. B. *et al.* (2009) 'Experience leaves a lasting structural trace in cortical circuits', *Nature*, 457(7227), pp. 313–317.

Holtmaat, A. and Svoboda, K. (2009) 'Experience-dependent structural synaptic plasticity in the mammalian brain', *Nature reviews. Neuroscience*, 10(9), pp. 647–658.

Huang, L. *et al.* (2021) 'Relationship between simultaneously recorded spiking activity and fluorescence signal in GCaMP6 transgenic mice', *eLife*, 10. doi: 10.7554/eLife.51675.

Lee, S., Park, J. and Smirnakis, S. M. (2019) 'Internal Gain Modulations, But Not Changes in Stimulus Contrast, Preserve the Neural Code', *The Journal of neuroscience: the official journal of the Society for Neuroscience*, 39(9), pp. 1671–1687.

- Marks, T. D. and Goard, M. J. (2020) 'Stimulus-dependent representational drift in primary visual cortex', *bioRxiv*. Available at: <https://www.biorxiv.org/content/10.1101/2020.12.10.420620v1.abstract>.
- Ma, Z. *et al.* (2020) 'Stability of motor cortex network states during learning-associated neural reorganizations', *Journal of neurophysiology*, 124(5), pp. 1327–1342.
- Niell, C. M. and Stryker, M. P. (2010) 'Modulation of visual responses by behavioral state in mouse visual cortex', *Neuron*, 65(4), pp. 472–479.
- Peters, A. J., Chen, S. X. and Komiyama, T. (2014) 'Emergence of reproducible spatiotemporal activity during motor learning', *Nature*, pp. 263–267. doi: 10.1038/nature13235.
- Stringer, C., Michaelos, M. and Pachitariu, M. (2019) 'High precision coding in visual cortex', *Cold Spring Harbor Laboratory*. doi: 10.1101/679324.
- Theis, L. *et al.* (2016) 'Benchmarking Spike Rate Inference in Population Calcium Imaging', *Neuron*, pp. 471–482. doi: 10.1016/j.neuron.2016.04.014.
- Vinck, M. *et al.* (2015) 'Arousal and locomotion make distinct contributions to cortical activity patterns and visual encoding', *Neuron*, 86(3), pp. 740–754.
- Wei, Z. *et al.* (no date) 'A comparison of neuronal population dynamics measured with calcium imaging and electrophysiology'. doi: 10.1101/840686.
- Zhang, Z. *et al.* (2018) 'Closed-loop all-optical interrogation of neural circuits in vivo', *Nature methods*, 15(12), pp. 1037–1040.
- Ziv, Y. *et al.* (2013) 'Long-term dynamics of CA1 hippocampal place codes', *Nature neuroscience*, 16(3), pp. 264–266.

REVIEWER COMMENTS

Reviewer #1 (Remarks to the Author):

The revised manuscript has clarified the methodological details of the conducted analyses. The new analysis in Supplemental Figure 5c-d, examining ability of SPUD to generalize across weeks is a great addition. However, my enthusiasm for the manuscript remains low as it seems to argue against a straw man idea of visual cortex responses that change so dramatically from week to week as to be unrecognisable. While it is clear that V1 responses change from week to week, the straightforward analysis in Figure 1c shows that the correlation of trial-average responses of individual neurons remains >0.5 , presumably with many individual neurons showing more stable responses. Moreover, this estimate likely represents a lower bound as it does not take into account technical sources of noise.

My main concern remains that the study's conclusions stem to a large a degree from the periodic structure of the stimulus and well-established properties of visual cortical responses. In the rebuttal, the authors respond that "stable representation of the time in the natural movie requires not only some neurons display some consistency in their responses over sessions, but also that 1) different groups of neurons be activated at different timepoints in the movie and 2) the activation of episodic activity tile across the entire duration of the movie (no silent period during the movie)." Both of these conditions are clearly satisfied in the visual cortex as long as 1) different segments of the movie are sufficiently distinct and 2) sufficient numbers of neurons are recorded to include neurons active at each timepoint.

The analysis in Figure 7, which uses a shuffling approach to disrupt coordination between neuronal responses is potentially interesting but it is not explored further. A likely interpretation is that shuffling the trial factors disrupts the structure of correlated variability of neuronal responses, which contributes to the identified low-dimensional representation. However, it is not clear what new insights on the role of correlated variability in representation of information by neuronal populations (which has been extensively studied in theoretical and experimental work not discussed in the manuscript) are revealed by the present analysis.

I do not want to be the sole holdout blocking publication, but I have to admit that I am still unsure about what I have learned from this study. While the approach is novel and interesting, it could have perhaps found its place as part of a larger paper examining the stability of V1 responses. On its own, I am still not convinced that it constitutes a significant advance.

Reviewer #2 (Remarks to the Author):

Thank you for thoroughly addressing all my comments. It was interesting to see the changes (or lack of changes) in the neuropil across weeks. Is there any reason not to include this figure as a supplemental figure with some quantification across neurons?

Reviewer #3 (Remarks to the Author):

The authors have partially addressed the comments raised.

I would have liked to see a more detailed neuron to neuron gain analysis (as opposed to the analysis of episodic population results), as well as a more detailed analysis of how behavioral parameters may or may not influence the results. The authors discuss the latter issue and expect that changes in behavioral parameters will result in an overall shift along the radial direction across the ring manifold, but they do not provide data to demonstrate if this is or is not the case. If these data are not available I recommend this is clearly stated and added as a possible caveat in the discussion.

Having said that, I believe that the authors contribution is significant, but would recommend addressing more thoroughly potential limitations of the study in the discussion, particularly as it pertains to the two points raised above.

I would also recommend additional polishing of the English language in the manuscript. Significant editorial corrections still need to be made to correct the English grammar in several places and ensure the text is easier to understand.

We thank the reviewers for their suggestions and insightful comments. Changes to the manuscript made in response to reviewer comments are marked in red. For the reviewers' convenience, we respond (in blue text) to each comment individually.

Reviewer #1 (Remarks to the Author):

The revised manuscript has clarified the methodological details of the conducted analyses. The new analysis in Supplemental Figure 5c-d, examining the ability of SPUD to generalize across weeks is a great addition. However, my enthusiasm for the manuscript remains low as it seems to argue against a straw man idea of visual cortex responses that change so dramatically from week to week as to be unrecognisable. While it is clear that V1 responses change from week to week, the straightforward analysis in Figure 1c shows that the correlation of trial-average responses of individual neurons remains >0.5 , presumably with many individual neurons showing more stable responses. Moreover, this estimate likely represents a lower bound as it does not take into account technical sources of noise.

The statement we tried to make with Fig 1c is that we observed gradually decreasing similarity over weeks using the first week of recording as the reference. Consequently, on average, individual neuronal responses gradually drifted away from its original response at the first week. However, as suggested by the reviewer, the amplitude of similarity showed that the drift we see in the visual cortex is not as big as the drift observed in other brain areas, such as the primary olfactory cortex (Schoonover et al. 2021).

My main concern remains that the study's conclusions stem to a large degree from the periodic structure of the stimulus and well-established properties of visual cortical responses. In the rebuttal, the authors respond that "stable representation of the time in the natural movie requires not only some neurons display some consistency in their responses over sessions, but also that 1) different groups of neurons be activated at different timepoints in the movie and 2) the activation of episodic activity tile across the entire duration of the movie (no silent period during the movie)." Both of these conditions are clearly satisfied in the visual cortex as long as 1) different segments of the movie are sufficiently distinct and 2) sufficient numbers of neurons are recorded to include neurons active at each timepoint.

Given that the long-term stability of visual cortical responses was not thoroughly studied before, the stable representation of visual stimuli is not guaranteed from the periodic structure of the stimulus and previously studied within-session properties of V1 responses. For example, motivated by the previous studies about long-term stability of neural activity in PPC (Driscoll et al. 2017), one could have expected to see the timing of spiking episodes in V1 neurons shifted across weeks, thus leading to unstable representation of time in the movie. Furthermore, recent study in the primary olfactory cortex (Schoonover et al. 2021) demonstrated that it is possible to have unstable stimulus representation during a relatively short period (~32 days) for primary sensory cortices. Our study clarifies that we still see stable representation of time in the natural movie in V1, given unstable gain changes in the spiking episodes.

The analysis in Figure 7, which uses a shuffling approach to disrupt coordination between neuronal responses is potentially interesting but it is not explored further. A likely interpretation is that shuffling the trial factors disrupts the structure of correlated variability of neuronal responses, which contributes to the identified low-dimensional representation. However, it is not clear what new insights on the role of correlated variability in representation of information by neuronal populations (which has been extensively studied in theoretical and experimental work not discussed in the manuscript) are revealed by the present analysis.

We agree with the reviewer that it would be interesting to explore further. Yet, we think it may be difficult to directly compare analysis in Fig 7 to the study of correlated variability for the following reasons. First, the study of correlated variability assumes that the neural activity is fluctuating around trial-averaged activity stationary over trials. However, in our case, we have the drifting trial-averaged activity over weeks. By shuffling the trial factors, we actually changed the trial-averaged activity for each week. Second, correlated variability described the coordination between neurons, while in our case, the shuffling disrupted the coordination between spiking episodes that are shared across neurons.

I do not want to be the sole holdout blocking publication, but I have to admit that I am still unsure about what I have learned from this study. While the approach is novel and interesting, it could have perhaps found its place as part of a larger paper examining the stability of V1 responses. On its own, I am still not convinced that it constitutes a significant advance.

Reviewer #2 (Remarks to the Author):

Thank you for thoroughly addressing all my comments. It was interesting to see the changes (or lack of changes) in the neuropil across weeks. Is there any reason not to include this figure as a supplemental figure with some quantification across neurons?

Motivated by the reviewer's comment, we decided to include the figure with Neuropil data as a supplemental figure (see Supplemental figure 8).

Reviewer #3 (Remarks to the Author):

The authors have partially addressed the comments raised.

I would have liked to see a more detailed neuron to neuron gain analysis (as opposed to the analysis of episodic population results), as well as a more detailed analysis of how behavioral parameters may or may not influence the results. The authors discuss the latter issue and expect that changes in behavioral parameters will result in an overall shift along the radial direction across the ring manifold, but they do not provide data to demonstrate if this is or is not the case. If these data are not available I recommend this is clearly stated and added as a possible caveat in the discussion.

Having said that, I believe that the author's contribution is significant, but would recommend addressing more thoroughly potential limitations of the study in the discussion, particularly as it pertains to the two points raised above.

We thank the reviewer for the suggestion and comment.

We defined gain modulation in the unit of spiking episodes instead of individual neurons, because we observed the gain modulation for different spiking episodes within the same neurons to be highly heterogeneous, i.e. for the same neuron, there exist some episodes with increasing gain and some episodes with decreasing gain (see Figure 2e, f). Thus, the neuron to neuron gain analysis might not be as useful as episode to episode gain analysis in our case.

The behavioral data for locomotion is not available for our study. However, in the Figure S10-11 of our co-submitted manuscript (Marks and Goard 2020), we discussed the statistics of eye movement and pupil size and their correlation with the neural data. And we only observe a weak correlation between pupil area and the radius of the manifold (see figure below).

Radius is weakly correlated with the pupil area shown for 1 imaging field. Y axis shows the radius of the projected instantaneous population activity, X axis shows the pupil area (quantified as the number of pixels in the pupil).

Motivated by the reviewer's comment, we add the following sentence in the discussion to address the limitation of our study (see Line 434) : "Furthermore, although we checked the limited impact of eye movement and pupil size on the stability of neuronal responses in an earlier work (Marks and Goard 2020), how behavioral variables would affect the neural trajectories remains to be explored in the future."

I would also recommend additional polishing of the English language in the manuscript. Significant editorial corrections still need to be made to correct the English grammar in several places and ensure the text is easier to understand.

Thanks for the advice, we will follow the editor's suggestions on polishing the language in the manuscript.

Reference

- Driscoll, Laura N., Noah L. Pettit, Matthias Minderer, Selmaan N. Chettih, and Christopher D. Harvey. 2017. "Dynamic Reorganization of Neuronal Activity Patterns in Parietal Cortex." *Cell* 170 (5): 986–99.e16.
- Marks, T. D., and M. J. Goard. 2020. "Stimulus-Dependent Representational Drift in Primary Visual Cortex." *bioRxiv*.
<https://www.biorxiv.org/content/10.1101/2020.12.10.420620v1.abstract>.
- Schoonover, Carl E., Sarah N. Ohashi, Richard Axel, and Andrew J. P. Fink. 2021. "Representational Drift in Primary Olfactory Cortex." *Nature* 594 (7864): 541–46.